# Superplume mantle tracked isotopically the length of Africa from the Indian Ocean to the Red Sea

John M. O'Connor [1,2,3*], Wilfried Jokat [1,4], Marcel Regelous[2], Klaudia F. Kuiper[3], Daniel P. Miggins[5] & Anthony A.P. Koppers[5]

Seismological findings show a complex scenario of plume upwellings from a deep thermo-chemical anomaly (superplume) beneath the East African Rift System (EARS). It is unclear if these geophysical observations represent a true picture of the superplume and its influence on magmatism along the EARS. Thus, it is essential to find a geochemical tracer to establish where upwellings are connected to the deep-seated thermo-chemical anomaly. Here we identify a unique non-volatile superplume isotopic signature ('C') in the youngest (after 10 Ma) phase of widespread EARS rift-related magmatism where it extends into the Indian Ocean and the Red Sea. This is the first sound evidence that the superplume influences the EARS far from the low seismic velocities in the magma-rich northern half. Our finding shows for the first time that superplume mantle exists beneath the rift the length of Africa from the Red Sea to the Indian Ocean offshore southern Mozambique.

[1] Alfred Wegener Institute Helmholtz Centre for Polar and Marine Research, Am Handelshafen 12, 27570 Bremerhaven, Germany. [2] GeoZentrum Nordbayern, Friedrich-Alexander-Universität Erlangen-Nürnberg, Schlossgarten 5, 91054 Erlangen, Germany. [3] Faculty of Science, Vrije University Amsterdam, De Boelelaan 1085, 1081 HV Amsterdam, Netherlands. [4] University of Bremen, Fachbereich 5, 28359 Bremen, Germany. [5] College of Earth, Ocean, and Atmospheric Sciences, Oregon State University, Corvallis, OR 97331-5503, USA. *email: j.m.oconnor@vu.nl

Extensive volcanism and tectonic activity in the East African Rift system (EARS) are regarded as the classic example of present-day plume-related rifting and continental breakup[1–3]. Rifting began in southern Ethiopia at approximately 45 Ma[4] with volcanism in northern Ethiopia and Yemen starting ~30 Ma[5]. This older part of the EARS stretches for >2000 km, originating from the Afar triple junction and traversing the Ethiopia Dome southward along the Main Ethiopian Rift (MER) before bifurcating into the Kenyan and Western rifts (Fig. 1).

Global and continental scale tomographic models show strong low-velocity regions in the mantle under East Africa (e.g. refs. [5–9]). One scenario proposed to explain these markedly lower mantle seismic velocities is the upwelling of a large, continuous-mantle plume (the African Superplume), originating at the core–mantle boundary[10,11]. The thermochemical nature of the seismic anomaly (superplume) in the lower mantle suggests a dense chemical layer within a buoyant upward-flowing thermal structure[12].

The topography of the magma-rich northern half of the EARS is dominated by two prominent plateaus—the Ethiopia and East African domes—which are transected by the EARS, the Red Sea and the Gulf of Aden. These domes are compatible with dynamic support by plume upwellings from the lower mantle rising beneath the continental lithosphere (e.g. refs. [10,11]), but it is unclear if one or more plumes support these high plateaus[2,6–9,13]. Moreover, recent tomographic models of the Afro-Arabian mantle reveal a laterally continuous, low-velocity region in the upper mantle beneath all of eastern Africa and western Arabia extending to depths of ~500–700 km[9–11,14].

The relation between these seismically imaged upwellings and large-volume EARS magmatism is often considered in the context of the well cited explanation that the superplume is a thermal phenomenon. In this scenario large-volume melting occurs where hot plume material flows below areas of thinned lithosphere and the $T_P$ is above the solidus. Continental lithosphere is inherently heterogeneous, specifically in terms of thickness. In the case of Africa, the Archean cratonic roots are deep so melts could only be generated beneath pre-existing thinner Karoo and Palaeogene orogenic belts adjacent to cratons[5]. The scattered distribution of rift-related EARS magmatism can therefore be explained by a variable lithospheric topography at depths of 100–150 km, which channels hot plume material into streams and pools[2,15]. However, a new generation of petrologic models[16] show that while the $T_P$ of the East African mantle is characterized by elevated temperatures (+140 to +170 °C) consistent with a buoyant superplume, it is toward the cooler end of the spectrum for Large Igneous Provinces (LIPs)[17]. Thus, factors other than elevated temperature must be contributing to large-volume melting and the exceptionally slow seismic velocity under East Africa. Consequently, there is much uncertainty about how well seismic data and large-volume melting captures the scale and dynamics of the superplume at depth and its interconnectivity with the upper mantle/EARS.

A way of resolving this debate would be to find a unique isotopic fingerprint for linking EARS lavas to a single deep-seated superplume. A number of such plume isotopic signatures have been inferred by interpolation of arrays in Sr–Nd–Pb isotopic space for lavas from the magma-rich northern EARS (e.g. refs. [13,18–21]), but it is unknown which, if any, of these isotopic signatures might be a unique tracer for the single superplume largely because of contamination/dilution by large-volume melting of the subcontinental lithospheric mantle. While rare gas isotopes are more successful in overcoming the problem of contamination/dilution[22,23], they cannot be used as a robust superplume tracer because they do not define a distinct isotopic signature and are not coupled to the solid superplume mantle.

The southern half of the EARS continues south of the Western Rift as far as the Davie Ridge where it trends NNE–SSW across the Mozambique Channel[24] and as far east as the Comoros-Mayotte and Madagascar system (Fig. 1). There is no seismic or geochemical evidence suggesting a connection between the African superplume and this magma-poor southern half of the EARS. But here we identify a unique non-volatile isotopic signature ('C') in the youngest (after 10 Ma) phase of widespread EARS rift-related magmatism where it extends into the Indian Ocean and the Red Sea. These offshore rock samples provide the first robust (isotopic) evidence that superplume material is present in the magma-poor southern half of the EARS, far from the very-low seismic velocities beneath the magma-rich northern half. This directly measured non-volatile superplume isotopic signature tracks for the first time superplume mantle the length of Africa from the Indian Ocean to the Red Sea. This more-or-less homogeneous isotopic composition is consistent with the inference from seismic mantle images of a single deep-seated superplume that is modifying the EARS mantle not only in northeast Africa but the length of Africa from the Red Sea to southern

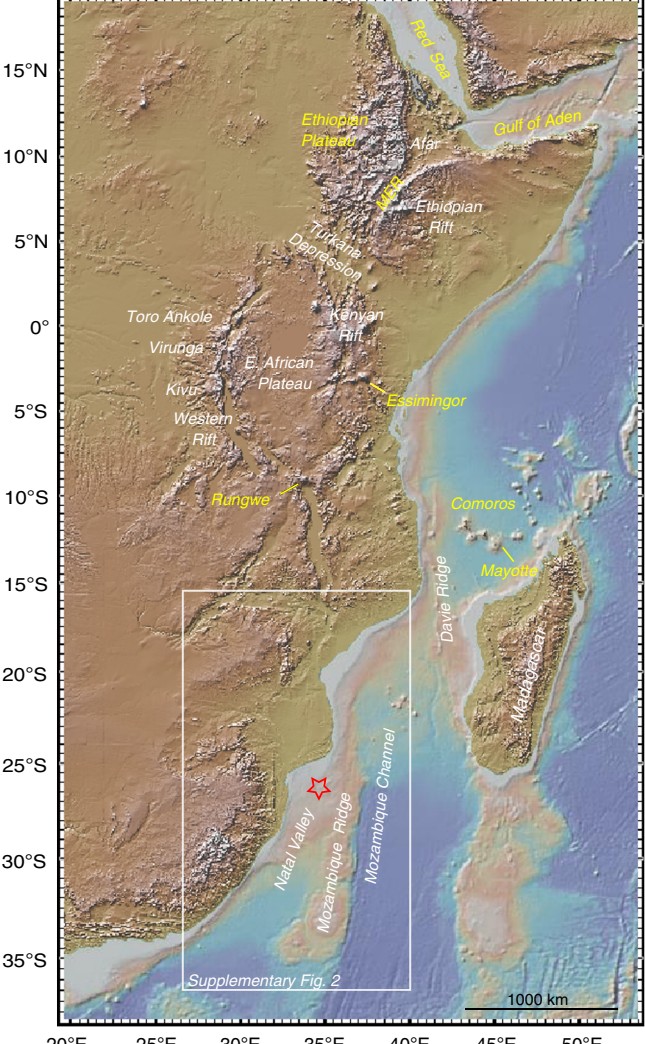

**Fig. 1** Topography and bathymetry map of the East African Rift System (NOAA globe 1 km). Map prepared using GeoMapApp (www.geomapapp.org). MER = Main Ethopian Rift. Red star shows location of the 7 Ma dome on the Mozambique Ridge. Yellow labels indicate isotopically dated volcanoes and plateaus associated with the most recent phase of rift-related magmatism discussed in the text.

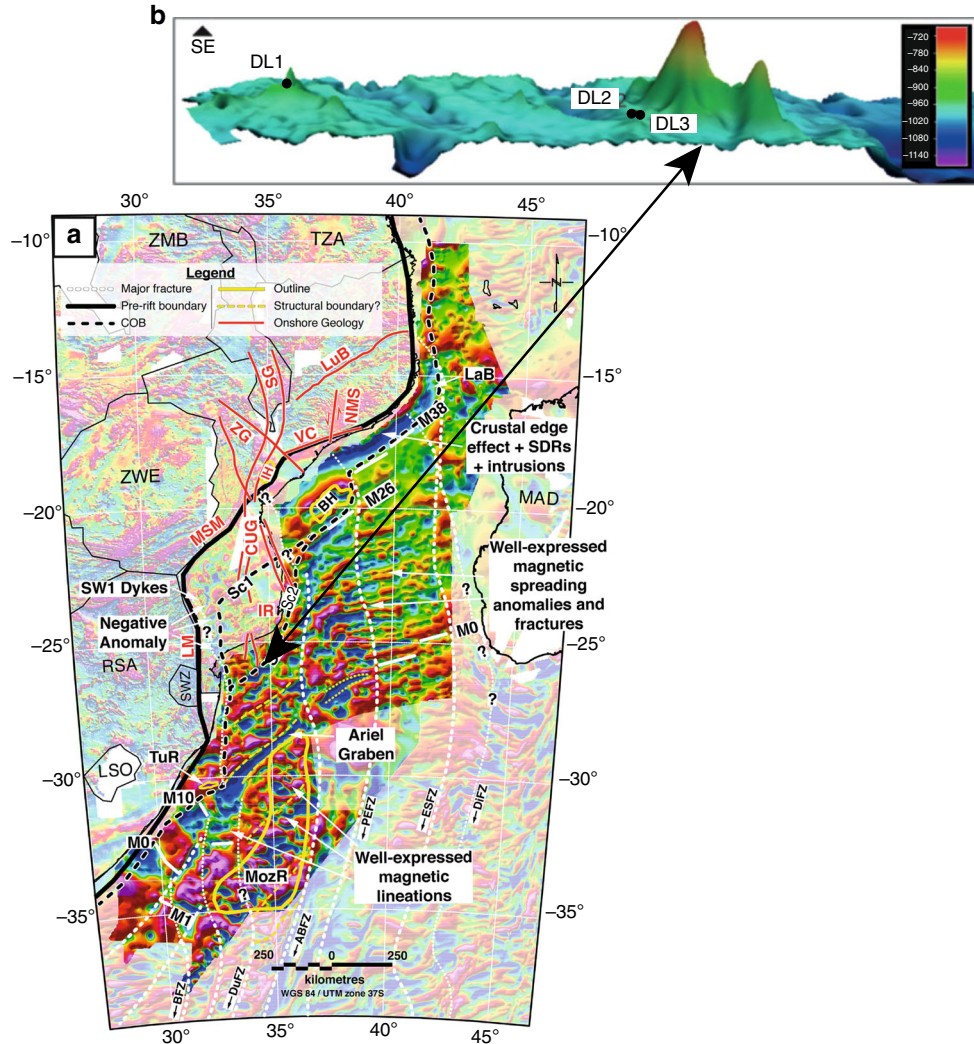

**Fig. 2** Young volcanism on the Mozambique Ridge. **a** Gridded anomaly map of the total magnetic field intensity (TMI) in the Mozambique Basin from ref. [31] showing that the entire Mozambique Ridge is characterized by well-expressed magnetic lineations and the major fractures and structural boundaries in the Mozambique Basin, relevant to Gondwana breakup, and selected major tectonic structures onshore SE-Africa. The locations of the magnetic spreading anomalies in the Mozambique Basin are taken from refs. [27,30]. See ref. [31] for further details and abbreviations of tectonic features. Continent-Ocean boundary (COB) inferred from scenarios 1 and 2 as proposed in ref. [30]. **b** Perspective map view showing the sampled dome on the Mozambique Ridge that is overall about 3 km east–west and 1.2 km in diameter and TV-grab sites DL2 and DL3. Figure from ref. [33].

Mozambique. Finally, we offer some broader perspectives about how our finding raises some interesting questions for future lines of research.

## Results and discussion

**Volcanism on the northern Mozambique Ridge**. The massive Jurassic/Early Cretaceous Mozambique Ridge is considered to be an LIP that formed during early Gondwana dispersal in the African–Antarctic corridor[25–31]. The northern Natal Valley was emplaced between M26r to M18n (157.1–144 Ma) and the entire Mozambique Ridge between M18n to M6n (144–131.7 Ma)[27,30,31].

Recently collected multibeam swath bathymetry and seismic profiles from the Natal Valley and Mozambique Ridge reveal anomalous seafloor dome-like volcanic structures, which are interpreted in terms of Miocene to recent magmatism along a southward propagation of the western branch of the East African Rift System[24,32–36] (Fig. 1). In the Natal Valley the domes are 25–31 km long and 16–18 km wide and rise some 400 m above the sedimentary deposits[35] (Fig. 2). Detailed swath data of

seafloor domes on the Mozambique Ridge[33,34] guided TV-grab sampling of volcanic rock from a smaller topographic feature (~3 km east–west and 1.2 km in diameter) at the northern end of the ridge (for details see ref. [34]) (Fig. 2). Information about the samples used in this study is provided as Supplementary Table 1 and Supplementary Fig. 1.

**$^{40}Ar/^{39}Ar$ geochronology**. We report here $^{40}Ar/^{39}Ar$ incremental heating isotopic ages, measured on acid-leached groundmass separates for three TV-grab samples recovered during the SO230 expedition (MW14DL2-2, MW14DL2-3 and MW14DL3-5) from the volcanic dome located at the northern end of the submarine Mozambique Ridge. $^{40}Ar/^{39}Ar$ incremental heating ages are summarized in Table 1 and Supplementary Fig. 2. Replicated (Vrije University Amsterdam and Oregon State University) robust ages for three samples from two different TV-grab stations demonstrate that the sampled dome volcanism is ≥6.9 ± 0.03 Ma. $^{40}Ar/^{39}Ar$ results, plateau ages and K/Ca spectra are provided in summary as Supplementary Data 1 and in full in Supplementary Data 2.

**Table 1 Results of $^{40}$Ar/$^{39}$Ar groundmass incremental heating experiments for SO230 samples.**

| Sample information | | Plateau | | | | | | Inverse isochron | |
|---|---|---|---|---|---|---|---|---|---|
| Experiment | Sample | Age ± 2$\sigma$ (i) | $^{39}$Ar (%) | K/Ca | MSWD | n | N | Age ± 2$\sigma$ (i) | $^{40}$Ar/$^{36}$Ar intercept ± 2$\sigma$ |
| [a]17D17844 | MW14DL3-5 | 6.90 ± 0.02 Ma | 72% | 0.357 | 1.35 | 12 | 24 | 6.90 ± 0.02 Ma | 287.03 ± 7.33 |
| [b]VU107-J3_1 | MW14DL3-5 | 6.94 ± 0.03 Ma | 73% | 0.325 | 1.87 | 8 | 15 | 6.89 ± 0.04 Ma | 316.22 ± 10.88 |
| [a]17D17979 | MW14DL2-3 | 6.91 ± 0.02 Ma | 52% | 0.267 | 3.26 | 7 | 24 | 6.89 ± 0.05 Ma | 315.46 ± 31.97 |
| [b]VU107-J2_1 | MW14DL2-3 | 6.99 ± 0.03 Ma | 63% | 0.326 | 2.07 | 6 | 15 | 6.92 ± 0.06 Ma | 321.16 ± 15.79 |
| [a]17D18018 | MW14DL2-2 | 6.97 ± 0.02 Ma | 57% | 0.173 | 0.40 | 11 | 24 | 6.97 ± 0.03 Ma | 354.12 ± 7.97 |
| [b]VU107-J1_1 | MW14DL2-2 | *7.26 ± 0.12 Ma* | *43%* | *0.168* | *7.94* | *4* | *15* | *6.87 ± 0.88 Ma* | *332.07 ± 74.98* |

Italics are for a rejected age
[a]200–180 μm; 1 N HCl (60 min); 6 N HCl (60 min); 1 N HNO$_3$ (60 min); 3 N HNO$_3$ (60 min); Mill-Q (60 min)
[b]355–200 μm; 1 N HNO$_3$ (120 min); Mill-Q (60 min)

**Geochemistry**. Based on silica and total alkali content samples DL2-2 and DL2-3 are basanites and DL3-5 is transitional between a basanite and an alkali basalt (Fig. 3). Trace element data show that the Mozambique Ridge samples are typical intraplate oceanic alkali basalts (Figs. 3 and 4) and have relatively radiogenic Pb isotope ratios ($^{206}$Pb/$^{204}$Pb = 19.52–19.54) and comparatively depleted Sr ($^{86}$Sr/$^{87}$Sr = 0.7031) and Nd ($^{143}$Nd/$^{144}$Nd = 0.5128) signatures (Fig. 5). Geochemical data are provided in Supplementary Data 3.

**EARS magmatism since the Late Miocene**. We consider first how an age of ~7 Ma for volcanism on the Mozambique Ridge compares with published ages for the EARS and intraplate volcanism in the Indian Ocean (Fig. 1). The Comoro Islands are associated with the seaward extension of the EARS and have recently been linked with a deep-seated plume[9,24,36]. K–Ar ages for the oldest island of Mayotte show that volcanism started roughly 8 ± 1 Ma and continued to at least as recent as 2–1 Ma[37,38]. Grande Comore is still active and K–Ar ages for Grande Comore, Moheli, and Anjouan are all younger than 8 ± 1 Ma[37,38]. Thus, K–Ar dating of the Comoro Islands suggest that subaerial volcanism initiated roughly around the same time as the dome on the Mozambique Ridge.

A $^{40}$Ar/$^{39}$Ar study of the Rungwe Volcanic Province (RVP) at the southern tip of the Western Rift (Fig. 1) yielded ages from 8.5 to 5.7 Ma[39], consistent with prior results, supporting an eruptive episode concurrent with tectonic activity on the Malawi and Rukwa border faults[40,41]. Three additional samples yield ages from 18.5 to 17.6 Ma, consistent with the 18.6 ± 1.0 Ma age obtained previously (see ref. 39 and reference therein). These older ages are spatially limited to phonolite domes that predate the current tectonic extensional structure. No Rungwe samples dated yet can be the source of the of 26 Ma carbonatitic tuffs in the nearby Rukwa area[42]. A K–Ar age determination from Essimingor volcano on the southern tip of the eastern branch (Fig. 1) suggest an age ≥8 Ma[43], whereas $^{40}$Ar/$^{39}$Ar ages indicate that volcanism started at ~6 Ma and has continued episodically to the present[44,45].

In the Ethiopian Plateau the oldest volcanic rocks have been dated at ~45 Ma in southern Ethiopia[2,4,46,47]. The most voluminous eruptive stage is represented by flood and shield basalts in the Ethiopian plateau made up of several distinct volcanic centres ranging from a thick sequence of ~30 Ma flood basalts overlain by a 30 Ma shield volcano. In the centre of the province are two ~22 Ma shield volcanoes[47–49] and younger shields developed ~11 Ma[48]. Between 12 and 10 Ma, the southern Red Sea margin propagated southward as the MER propagated NE, effectively linking the southern Red Sea and Ethiopian rifts, and forming a triple junction for the first time, possibly in

response to a (global) plate reorganization[49]. In the Gulf of Aden ages of off-axis seamounts point to a volcanic event at ~10 Ma that subsequently became more widespread roughly 6 Ma ago[50].

In summary, the age of volcanism on the Mozambique Ridge and the Comoro Islands is similar to the overall timing of Late Miocene rift-related magmatism in the magma-rich northern half of the EARS and extending into the Red Sea-Gulf of Aden. Young volcanism on the Mozambique Ridge represents the southernmost known EARS-related magmatism, and extends the EARS volcanic province more than 2000 km to the south of Rungwe (Fig. 1).

**EARS isotopic signatures**. Sr–Nd–Pb isotopic fingerprints for plume mantle under the EARS were previously determined from the general convergence of lavas in Sr–Nd–Pb isotope space (see ref. 21 and reference therein). Three candidate plume isotopic signatures are proposed for the EARS: plume 'C'[20], plume 'K'[18] and plume 'V'[21].

*Plume 'C'*: In the northern part of the EARS, recent volcanic and magmatic activity is largely confined to the tectonically active segments of the MER and the active spreading centres of the Red Sea and the Gulf of Aden[49]. Lavas from along and near the MER are predominantly transitional tholeiites with Th/Yb and Nb/Yb ratios similar to OIB and E-MORB (Fig. 4) and have less enriched and more homogeneous Sr, Nd and Pb isotope compositions than EARS lavas to the south (Fig. 5) (see e.g. refs. 13,20,51). A widely accepted explanation for the Sr–Nd–Pb isotopic signature of the MER is that it results from mixing between plume material and MORB mantle (DMM), and continental lithosphere (SCLM) (see e.g. refs. 20,52). Converging MER arrays in multi-isotope space define the composition of the plume endmember (centred about $^{87}$Sr/$^{86}$Sr = 0.7035, $^{143}$Nd/$^{144}$Nd = 0.51287 and $^{206}$Pb/$^{204}$Pb = 19.5 (ref. 20) (Fig. 5). This plume endmember is similar to the 'common' isotopic composition observed globally in oceanic basalts[20] known as 'C'[53]. A key finding of our study is that the 7 Ma basanites-alkaline basalts from the Mozambique Ridge express a pure plume 'C' isotopic signature (Fig. 5).

*Plume K*: South of the Ethiopian Rift the EARS branches into the Kenyan and Western rifts. Lavas from the Kenyan Rift converge in Sr–Nd space to a composition referred to as plume K ($^{87}$Sr/$^{86}$Sr 0.7030–0.7035 and $^{143}$Nd/$^{144}$Nd 0.5130–0.5127)[4,13,18], which is largely indistinguishable from plume 'C' (Fig. 5).

*Plume V*: The compositional convergence of Western Rift lavas is attributed to the presence of a common superplume V, with a limited range of compositions, that had been continuously metasomatizing the compositionally variable lithosphere beneath the Western Rift for the last 500–1000 Ma[21]. The $^{87}$Sr/$^{86}$Sr, $^{143}$Nd/$^{144}$Nd, and $^{206}$Pb/$^{204}$Pb composition of plume V is represented best by lavas from the Nyiragongo and Nyiramugira

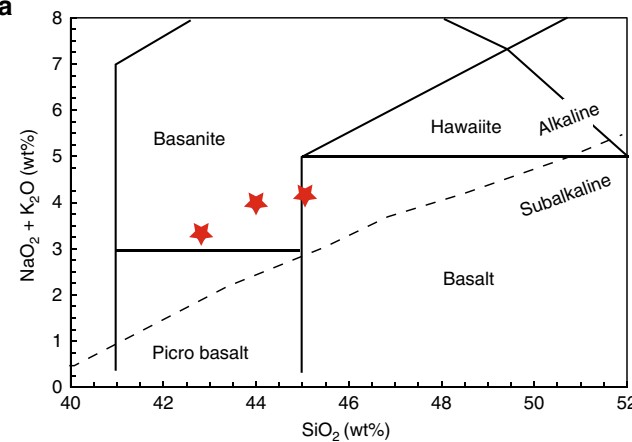

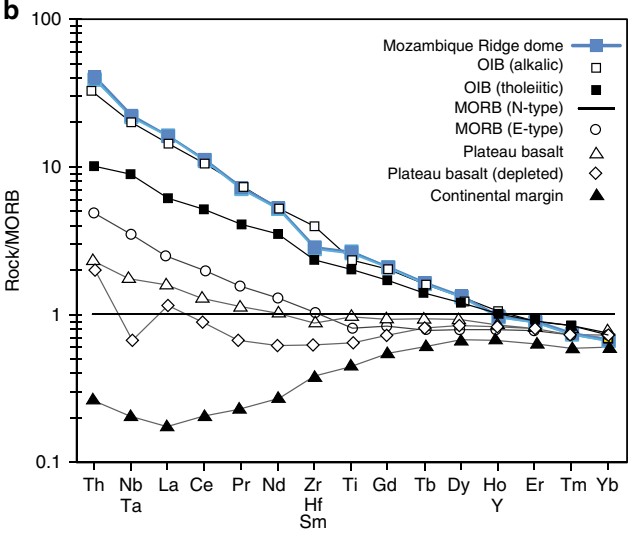

**Fig. 3** Major and trace element classification. **a** Total alkali vs. silica diagram. Based on silica and total alkali contents and the classification diagram of ref. [99] MW14DL2-2 and MW14DL2-3 are basanites and MW14DL3-5 is a transitional between a basanite and an alkali basalt. **b** MORB-normalized trace element concentration patterns of dome samples compared with the compositional range of oceanic basalt types in the present-day ocean basins from ref. [100] (see also for details of the following data sources). N-MORB (also the normalizing factor) and E-MORB data are from the compilation in Sun and McDonough[101]; intraplate islands (islands on ridge-distal oceanic lithosphere) are represented by alkalic OIB (the OIB average of Sun and McDonough)[101] and a tholeiitic basalt standard from Hawaii (BHVO1); the plateau basalt data are from the Ontong-Java Plateau in Mahoney et al.[102]; the depleted plateau basalt data is from Gorgona Island in Révillon et al.[103] Note that the data for the three dome samples are very similar and so plot on top of each other.

volcanoes (VVP)[54] (average $^{87}Sr/^{86}Sr$ 0.7045; $^{143}Nd/^{144}Nd$ 0.5127; $^{206}Pb/^{204}Pb$ 19.4)[21] (Fig. 5). Metasomatization of the lithospheric mantle at various times by this volatile-rich (primarily carbonatitic) plume material (ref. [21] and references therein and ref. [51]) is reflected in the exceptionally high level of trace element enrichment compared to the rest of the EARS in lavas from volcanic provinces in the Western Rift (Rungwe (RVP), Virunga (VVP), Toro Ankole, and Kivu) (Fig. 4)[55–58]. This process raised the $^3He/^4He$ ratio and led to almost complete hybridization of the trace elements, Sr–Nd, and, to a large extent, Pb isotope compositions[21,51]. Plume V $^{206}Pb/^{204}Pb$ (constrained

from the intersection of the Pb isotopic analyses from Western Rift volcanoes (e.g. refs. [59,60]) is strikingly similar to that of plume C (calculated from the intersection of Pb isotopic analyses from the MER[20]) (Fig. 5). An indication that Pb and Sr–Nd isotopes might be behaving differently is the positive correlation between He and Pb isotopes, but not Sr–Nd, for RVP and MER lavas[21]. This decoupling might be explained by 'outgassing' of abundant $CO_2$ from the superplume at depth acting as a carrier phase for the trace gases[21] (see discussion in a following section about other drivers of melting). RVP magmas are among those produced by the lowest degrees of melting along the Western Rift[21] (Fig. 4) suggesting that low-degree melting allows the Pb, if not the Sr–Nd, aspects of the common 'C' /V signature to bypass contamination by metasomatized SCLM.

In the following sections we discuss the evidence that a similar mechanism operating along the offshore extensions of the EARS in the Red Sea-Gulf of Aden, Southern Somali Basin and Indian Ocean results in lavas containing the Sr–Nd–Pb superplume isotopic signature without significant contamination or dilution by lithospheric sources.

**Red Sea-Gulf of Aden.** The Red Sea is a rift zone at a transient stage between continued development or failure where alkali volcanism dominates[52] (Supplementary Fig. 3a). The oldest known oceanic crust formed ~5 Ma ago at about 17°N, and since then oceanic spreading has developed progressively northwards[61]. A pure plume 'C' isotopic signature is evident in basalts from Ramad Seamount (~17°N) and in basanites from Jizan volcanic field located about 200 km to the east on the coastal plain[61] (Fig. 5). In the Gulf of Aden, where seafloor spreading is well established, plume C lavas occur at ~46°E[52]. An apparent mixing/dilution array between pure plume 'C' and DMM is indicated by alkali basalts from the Zubair (~15°N) and Hanish-Zukur (~14°N) archipelagos[52], lavas from the rift zone extending northwest of Ramad, and from the Sheba Rift in the Gulf of Aden (Figs. 4 and 5). Another source of contamination of the plume C signature is subcontinental mantle lithosphere, which has affected lavas from the intersection of the Sheba Ridge and the African margin (Tadjoura Trough, Gulf of Tadjoura and Asal Rift)[52] (Fig. 5). Schilling et al.[52] interpret the Sheba spreading ridge in the Gulf of Aden and its SW extension into the Tadjour Trough, Gulf of Tadjoura and the Asal Rift in terms of mixing between Pan-African continental lithosphere, the Afar mantle plume, and DMM. Moreover, these lavas lie on Sr–Nd–Pb isotopic arrays defined by Late Oligocene–Quaternary continental basalts from the southern end of the onshore Yemen Traps[62] and the Hamdan volcanic field to the north (15.30°–16°N)[61]. The Yemen Traps are associated with the Red Sea-Gulf of Aden rifting[62] and the source of this subcontinental mantle lithosphere is presumed to be related to the Pan-African orogenic events[5]. The correspondence between the Yemen Traps and the MER Sr–Nd–Pb isotopic arrays suggests that the contamination of the plume 'C'–DMM array by Pan-African continental lithosphere (e.g. refs. [13,20,51,52]) is confined to lavas erupted onshore.

On a global scale, variations in the thickness of the oceanic lithosphere exert a first-order control on the geochemistry of OIB despite other effects such as fertile mantle compositional heterogeneity. That is, the lithosphere thickness limits the mean extent and pressure (depth) of melting[63–65]. The Mozambique Ridge OIB and other EARS lavas have higher Nb/Yb and $TiO_2/Yb$ ratios than MORB due to the thicker lithospheric cap resuting in lower degrees of melting at greater depth with residual garnet (Fig. 4). The intensity of the garnet signature in OIB melts is also affected by mantle temperature; higher temperatures result in a deeper onset of melting and thus a greater garnet signature for a

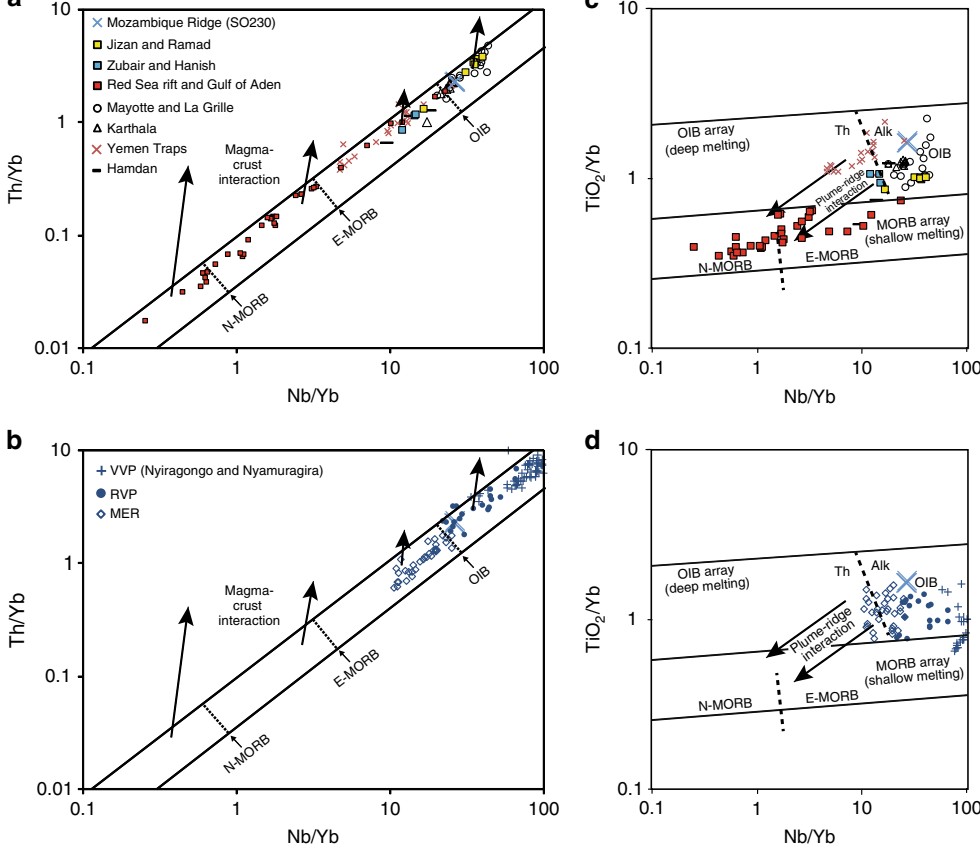

**Fig. 4** Trace elemental characteristics. Left panels show the Th–Nb proxy for crustal input[100]. **a** is for the offshore samples plus the Yemen Traps and Hamdan and **b** is for the MER, RVP and VVP volcanic provinces. Thorium enrichment is indicative of crustal contamination unrelated to magmatic processes. The Mozambique Ridge OIB samples and other lavas discussed in the text plot within the MORB–OIB array rather than above indicating no significant Th enrichment. Right panels show the Ti–Yb proxy for depth of melting[100]. **c** is for the offshore samples plus the Yemen Traps and Hamdan and **d** is for the MER, RVP and VVP volcanic provinces. Data sources are as follows: Main Ethiopian Rift (MER)[20], Rungwe Volcanic Province (RVP)[21], Virungu Volcanic Province (VVP)[21,54]; Tadjoura Trough, Tadjoura Rift, Asal Rift[52], Gulf of Aden, Red Sea rift (see compilation of ref. [104]); Ramad, Jizan, Zubair, Hanish, Hamdan[61]; Yemen Traps[62]; La Grille and Karthala (Grande Comore)[67–69,105] and Mayotte[66].

given lithosphere thickness. The high Th/Nb and Ti/Yb ratios (Fig. 4) show that the Mozambique Ridge lavas erupted on older, thickened lithosphere (high mean pressure and low mean extent of melting)[63–66]. Smaller degrees of melting beneath thicker lithosphere will tend to yield melts with higher incompatible trace element concentrations and a lower depleted mantle contribution, which are less susceptible to contamination.

**Southern Somali Basin**. The magma-poor southern half of the EARS extends offshore into the Somali Basin along the Comoros-Mayotte and Madagascar tectono-volcanic system[24,36] (Fig. 1). Grande Comore, the youngest of the Comoro islands, is composed of two active volcanoes, Karthala and La Grille, whereas volcanism on Mayotte started roughly 8 ± 1 Ma[37,38]. The overall isotopic variation of lavas from the islands of Grande Comore and Mayotte[66] is explained by mixing between a plume source with a nearly uniform isotopic and chemical composition and alkalic low-degree melts of metasomatized oceanic lithospheric mantle[67–69]. A few samples from Mayotte and to a lesser extent La Grille show a pure plume 'C' signature (Fig. 5). These lavas represent lower degree melts (Fig. 4) of the metasomatized oceanic lithospheric mantle[68]. In contrast, the non-plume 'C'-like Karthala alkali olivine basalts are the result of mixing between plume melts and higher degree melting of the metasomatized oceanic lithospheric mantle[68]. The lower degree La Grille melts might reflect a structural weakness in the 140-Ma-old oceanic

lithosphere of the southern Somali Basin[70] that allows easy migration of near-primary plume melts to the surface[68].

In summary, low-degree melts of pure superplume 'C' mantle generated under thickened oceanic lithosphere are transported rapidly to surface offshore via the EARS in the southern Somalia Basin. But in contrast to the Red Sea-Gulf of Aden and the Mozambique Ridge, long-term plume-related metasomatized oceanic lithospheric mantle represents an additional source of contamination.

**Mozambique Ridge**. The EARS extends offshore along the Comoros-Mayotte and Madagascar system and as far south as the Davie Ridge where it trends NNE–SSW across the Mozambique Channel and the Mozambique Ridge[24] (Supplementary Fig. 3). Seismic reflection and multibeam data[24] and references therein show that activity along a fault zone related to EARS is associated with the development of seamounts, dykes and lava flows in the Mozambique Channel[71,72] and the young volcanic domes on the Mozambique Ridge and in the northern Natal Valley[35] (Supplementary Fig. 3). These volcanic edifices most likely developed initially during the Mid-Miocene[71,72]. We show here that the 7 Ma basanites-alkaline basalts from the Mozambique Ridge volcanic domes express a pure plume 'C' isotopic signature identical to that of Jizan-Ramad lavas in the Red Sea (Fig. 5). Trace element ratios indicate that these lavas result from small degrees of melting in the garnet stability field (Fig. 4). We argue

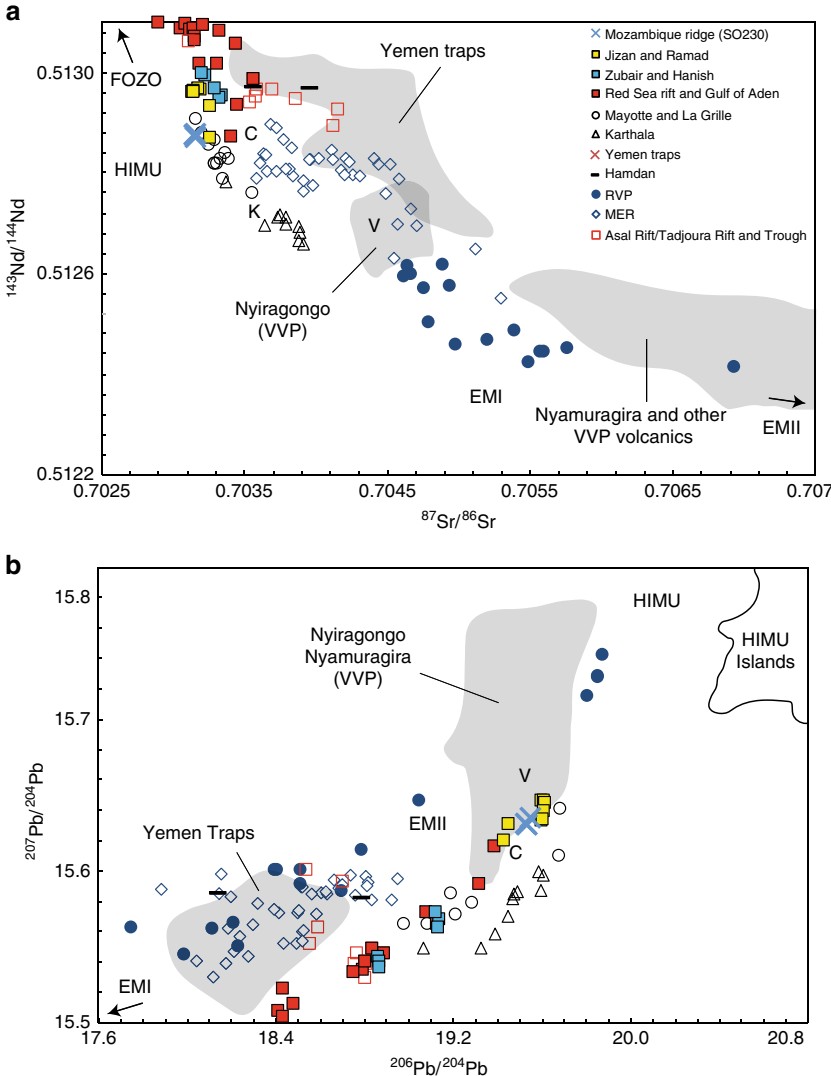

**Fig. 5** Radiogenic isotope characteristics. **a** $^{143}Nd/^{144}Nd$ vs. $^{87}Sr/^{86}Sr$ and **b** $^{206}Pb/^{204}Pb$ vs. $^{207}Pb/^{204}Pb$ for Mozambique Ridge OIB samples and lavas from the MER, RVP, Comoro Islands, Red Sea-Gulf of Aden. Lavas from Nyiragongo and Nyamuragira volcanoes in the Virunga Volcanic Province (VVP) and the Yemen Traps are shown as a grey shaded field. EMI, C, FOZO, EMII, BSE and HIMU are proposed mantle endmembers[106,107]. V, K, and C are the proposed Virunga (ref. [21], modified from ref. [54]), Kenyan[18] and Afar[20] mantle plumes, respectively. Note that V and C have the approximately the same $^{206}Pb/^{204}Pb$ values. The range of 'classic' HIMU islands is defined by the St. Helena and the Cook–Australs; data from the GEOROC database (http://georoc.mpch-mainz.gwdg.de/). Data for the Tadjoura Trough, Tadjoura Rift and Asal Rift from ref. [52]. Other data sources as in Fig. 4. Figure adapted from ref. [21].

that, as in the case of plume 'C' samples from the Red Sea-Gulf of Aden, the low mean extent of melting and absence of thick continental lithosphere minimizes dilution/contamination. This inference agrees with potential field data and magnetic lineations (Fig. 2) showing that the entire Natal Valley and Mozambique Ridge is floored by oceanic crust emplaced between 157.1–144 Ma and 144–131.7 Ma, respectively, with an anomalous thick oceanic layer 3 and/or significant magmatic underplate (refs. [26,31] and references therein) (Fig. 2).

In summary, we argue that as in the Red Sea and the Southern Somali Basin, the Mozambique Ridge lavas express the pure isotopic 'C' signature of the African superplume where complex tectonic settings such as thickened lithosphere and deep faults associated with the EARS facilitate rapid rise of low-volume melts to the surface without significant contamination/dilution by SCLM or DMM.

**Non-thermal drivers of melting**. We consider now various potential drivers of melting along the EARS that do not require a purely thermal superplume or its interaction with thinned/rapidly thinning lithosphere. Plate stretching in Ethiopia at different times during rift development seems to have generated large volumes of decompression melt in the asthenosphere leading to the lower seismic velocities observed in the underlying mantle (e.g. refs. [17,73,74]).

According to Rooney et al.[17], the compositional heterogeneity in the thermochemical superplume in the lower mantle[12] is likely due to recycled slab material that has been converted into eclogites and pyroxenites (e.g. ref. [75]). This suggestion is consistent with the apparent alignment of the EARS with the eastern side of the African Large-low-shear-wave-velocity provinces (LLSVPs) at the base of the mantle formed by the accumulation of piles of subducted ocean crust over Earth's history[76–78]. Slab mantle carries with it carbonate ($CO_2$) that can trigger large-volume deep melting in both the recycled crust and associated peridotite host[79], that likely contributes significantly to the low seismic wave speeds that characterize the East African mantle[17]. This notion is consistent with long-established

abundant $CO_2$ outgassing along the magma-rich northern half of the EARS[80–83]. Moreover, Lee et al.[83] find that significant volumes of $CO_2$, probably sourced from upper mantle or lower-crustal magma bodies, is transferred from upper mantle or lower-crustal magma bodies along the deep faults away from active volcanic centres (e.g., Rungwe) in the EARS. Subduction-related $CO_2$-assisted large-volume melting can explain the apparent contradiction between the relatively cool mantle temperatures (+140 to +170 °C) and markedly low seismic wave speeds associated with the magma-rich northern half of the EARS[17].

Superplume material in the upper mantle is also likely to occur as inherently more fusible mantle domains. For example, Rooney et al. [84] conclude that the widespread distribution of easily fusible lithospheric 'metasomes' within the EARS continental lithosphere mantle may facilitate magma generation without the need for substantial lithospheric thinning or elevated mantle potential temperatures. $CO_2$ might also be an important volatile phase contributing to small volume partial melting of small-scale fusible mantle domains[85] transferred from upper-mantle or lower-crustal magma bodies along the deep faults away from active volcanic centres in the EARS[83].

Small-scale convection is a little-understood potential driver of low-volume EARS melting[17]. Small upwellings emanating from the African superplume are inferred petrologically[48] and seismically[86], and, recent tomography models for east Africa reveal that the scale of upwellings under the magma-rich half of the EARS is smaller than expected for lower mantle plume sources[87,88].

In summary, large-volume melting along the continental EARS leads to contamination of the superplume signature by melting of (metasomatized) lithospheric sources. The Sr, Nd, and Pb isotopic superplume signature 'C' is seen most clearly in offshore lavas, where the EARS can act to transfer uncontaminated low-volume melts of more fusible (fertile ± $CO_2$), relatively cool ($T_P$ that ranges from ambient mantle to only 1490 °C), widely distributed plume material to the surface, possibly in association with small-scale plume convection.

**Rare gas vs. solid isotopic tracers**. Trace gas signatures such as the high $^3He/^4He$ isotopic ratios in lavas from the Ethiopia Rift and Rungwe are a potential tracer of deep-seated plumes[22]. Nevertheless, they cannot distinguish between the presence of a single superplume source located in the EARS mantle, or if the deeper superplume is connected directly to the shallow mantle via multiple (plume) upwellings or indirectly via multiple mantle plumes rising from one or more boundary layers[23]. Halldórsson et al.[23] show that combining He and Ne isotopes provide new insights into key aspects of the EARS mantle not possible using He isotopes alone. One finding is that there is a common mantle plume source beneath the magma-rich northern half of the EARS because hyperbolic mixing trajectories are compatible with admixture between an assumed PLUME and depleted (DMM) or subcontinental lithospheric (SCLM) mantle source endmembers. However, the He–Ne isotopic signature of their assumed PLUME source encompasses the range of many oceanic hot spots, such as Iceland[23]. Moreover, the authors note the extreme sensitivity of Ne isotopes to relatively small additions of the PLUME end-member to DMM and SCLM. A case in point is the hyperbolic mixing trajectory between PLUME and SCLM which provides the best fit to data from the Kenyan and Western rifts. But just a small amount of SCLM has a huge effect on $^4He/^3He$, such that the lavas are infered to have up to 80% of this component (see yellow symbols in their Fig. 3).

A further complication in extending the use of trace gases to map the distribution of superplume mantle is the $CO_2$-assisted

large-scale melt production contributing significantly to the low seismic wave speeds in the East African mantle[17]. The 'outgassing' of abundant $CO_2$ acts as a carrier phase for the trace gases, as is proposed to explain the decoupling of helium from Sr–Nd–Pb in the Hawaiian plume[89]. Thus, trace gases behave differently from the non-volatile plume tracers such as Sr–Nd–Pb and reflect therefore the spatial decoupling of $CO_2$ from the superplume. High $CO_2$ flux from the superplume will also contaminate/dilute the He–Ne plume signature with variable amounts of SCLM and DMM. If, for example, the large-scale $CO_2$ flux from the superplume differs between the magma-poor and magma-rich halves of the EARS then hyperbolic mixing trajectories will not predict the same/valid He–Ne superplume signature due to mixing with variable amounts of SCLM and DMM. In short, while hyperbolic mixing curves can provide a best estimate for the He–Ne isotopic signature of the African superplume it cannot be used as a robust isotopic tracer for superplume mantle. In contrast Sr, Nd and Pb isotopes can potentially measure directly a common plume isotopic signature, in part because the concentration contrast between plume and SCLM is far less significant. Thus, we report here the first robust, directly measured viable isotopic signature for tracking the influence of superplume in EARS lavas.

In summary, we have sampled basanites-alkaline basalts from one of the small domes on the Mozambique Ridge. These volcanic domes are located along the offshore southern extension of the EARS into the Indian Ocean on 144–157 Ma igneous crust. A ~7 Ma $^{40}Ar/^{39}Ar$ age for these OIB lavas shows that the widespread youngest (after 10 Ma) phase of EARS magmatism extends from the Red Sea as far south as the Mozambique Ridge. The OIB lavas have an Sr–Nd–Pb isotopic signature ('C'), which is also seen in lavas from the other ends of the EARS, at its offshore extension into the Red Sea and Somalia Basin. Beneath the continental EARS, the 'C' component is diluted due to contamination by large-volume melting of (metasomatized) lithospheric sources, but its presence can be inferred from variations in Sr–Nd–Pb isotope space. We conclude that the pure isotopic superplume signature 'C' is only sampled offshore where the EARS can act to transfer uncontaminated low-volume melts of more fusible (fertile ± $CO_2$), relatively cool (TP that ranges from ambient mantle to only 1490 °C), widely distributed plume material to the surface, possibly in association with small-scale plume convection. These offshore rock samples provide the first sound evidence that superplume material is present in the magma-poor southern half of the EARS, far from the very-low seismic velocities associated with the magma-rich northern half. We argue that because the EARS can tap pure plume 'C' lavas wherever the tectonic setting is appropriate then common 'plume 'C' superplume material must be widely distributed in the upper mantle under the entire EARS from the Mozambique Ridge to the Red Sea-Gulf of Aden. This finding implies that superplume material contributes to EARS magmatism not only in northeast Africa but along the length of the EARS from the Red Sea to southern Mozambique. We conclude therefore that we have isotopically tracked for the first time superplume mantle the length of Africa from the Indian Ocean to the Red Sea, which is consistent with seismic mantle images of a single deep-seated superplume with a more-or-less homogeneous isotopic composition. Our finding seems to offers some broader perspectives about the African superplume that raise some interesting questions for future lines of research as follows: Evidence that superplume mantle is associated with the magma-poor southern half of the EARS implies that the C mantle is reaching the EARS regardless of whether it is magma-rich or the existence of anomalously low mantle velocities (Fig. 6). This observation is consistent with the suggestion that the markedly low mantle velocities in the magma-

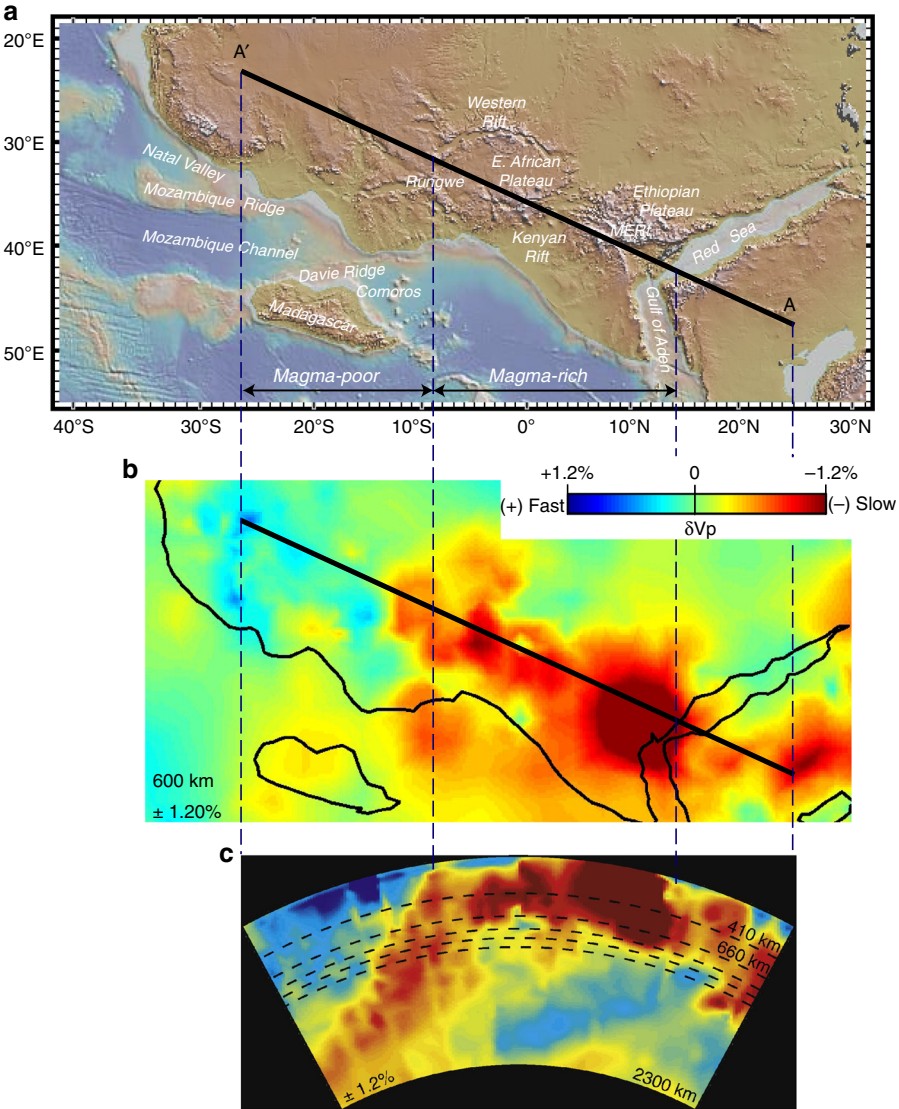

**Fig. 6** Schematic showing relation between seismic velocity and magmatism along the EARS. **a** Topography and bathymetry map showing the location of profile A–A′ from the P-wave tomography model of ref. [11] Other details as in Fig. 1. Figure adapted from ref. [11]. **b** P-wave velocity perturbations from the tomography model of ref. [11] are shown for a depth of 600 km in the mantle. Note the relation between the low-velocity region in the upper mantle extending to depths of ~500–700 km and the magma-rich northern half of the EARS. The ± values in each inset indicate the bounds of the colour scale used for the corresponding panel. **c** Cross-sectional view of the P-wave tomography model of ref. [11] along profile A–A highlighting the low-velocity features in the African and Arabian mantle. Figure adapted from ref. [11].

rich northern half of the EARS reflect the thermochemical structure of the superplume due to, for example, recycling of crust into the mantle carrying with it carbonate ($CO_2$) that can trigger deep melting. A further perspective is that it is increasingly more feasible to measure Sr–Nd–Pb signatures in melt inclusions in primary minerals[90] that might carry the pure superplume 'C' signature to the surface. These primary melts lock in the 'C' signature before it can be overwhelmed by contamination from large-volume melting of lithospheric sources. Thus, in due course the superplume 'C' fingerprint could be used to track superplume mantle in contaminated continental EARS lavas.

## Methods

**Geochemistry**. Approximately 0.05 g of sample was accurately weighed into a Teflon beaker, and digested in 1 ml 15 M $HNO_3$ and 3 ml 12 M HF for 12 h in sealed beakers on a hotplate at 80 °C. After cooling, 0.2 ml of $HClO_4$ was added to the sample, and the solution evaporated to incipient dryness at 120 °C. Two millilitres of 15 M $HNO_3$ was added to the sample, and evaporated to near dryness, and this step was repeated twice before increasing the hotplate temperature to

150 °C and fuming off excess $HClO_4$. The sample was then redissolved in 4 ml 15 M $HNO_3$ and 4 ml $H_2O$, two drops of 12 M HF were added, and the sealed beakers left on a hotplate at 80 °C for 12 h. The samples were then placed in an ultrasonic bath for 30 min, before heating at 80 °C for another 12 h. At this stage, all samples were completely in solution. The sample solutions were then quantitatively transferred to 250 ml HDPE bottles and diluted to 200 g with MQ water to obtain a final solution of 2% $HNO_3$ + 0.002 M HF with a sample dilution factor of about 4000 and total dissolved solids of 250 µg/ml. All reagents used were distilled in Teflon stills, and diluted with MQ 18.2 Ohm water.

Trace element measurements were carried out at the GeoZentrum Nordbayern using a Thermo Scientific X-Series 2 quadrupole inductively coupled plasma mass spectrometer. Samples were introduced into the instrument through a Cetac Aridus 2 desolvating nebulizer system in order to reduce molecular interferences. An ESI SC-2 DX FAST autosampler was used to reduce washout times between samples. The instrument was tuned using a 5 ppb solution of Be, In and U; typical sensitivity for $^{238}U$ was $2 \times 10^6$ counts per second for a sample uptake rate of 50 µl/min. The Ce/CeO ratio was typically >4500, and thus corrections for interference of oxides of Ba and the light rare-earth elements on Eu and Gd were unnecessary. Before each measurement session, the instrument was calibrated using multielement solutions covering the relevant concentration range. A mixed Be, Rh, In and Bi solution (30, 10, 10, 5 ppb) was mixed with the sample online and these elements used as internal standards to correct for instrumental drift. Procedural blanks analysed

during this work were negligible for all elements measured. Trace element data for the rock standard BHVO-2 measured as an unknown are given in the table.

Whole-rock powders for Sr, Nd and Pb isotope analysis were leached prior to dissolution. Approximately 1 g of powder was leached with 6 M HCl at 100 °C for 8 h, changing the acid every 2 h, and the remaining residue washed repeatedly with MQ $H_2O$, and then dried. Plagioclase separates prepared for Ar–Ar analysis were rinsed with MQ $H_2O$. For Pb isotope analysis, approximately 70 mg of sample was weighed into a Teflon beaker, and digested overnight in sealed beakers at 70 °C in a mixture of 10 drops 15 M $HNO_3$ and 20 drops 14 M HF. All acids used were double-distilled, except for HF which was purchased as Suprapur grade. The resulting solution was evaporated to near dryness, treated with 15 drops 15 M $HNO_3$, evaporated to complete dryness, redissolved in 20 drops 6 M HCl, evaporated and finally taken up in 20 drops 1 M HCl. The samples were transferred to 2 ml centrifuge tubes, centrifuged and the solution loaded onto ion-exchange columns containing 0.1 ml Eichrom SrSpec resin. Most elements were washed off the resin with 3 ml 1 M HCl, and this fraction was collected for Sr and Nd. Pb was recovered with 2 ml 6 M HCl, and this solution evaporated, redissolved in 1 M HCl and passed a second time through the columns. The final solution was dissolved in 2% $HNO_3$ for mass spectrometry.

The 1 M HCl solution collected from the SrSpec resin was dried down and redissolved in 3.5 M $HNO_3$. The REE were separated using ion-exchange columns containing 0.1 ml Eichrom TRUSpec resin, positioned to drip into a second column containing 0.1 ml SrSpec resin, which retained the Sr. After washing with 1 ml 3.5 M $HNO_3$, the columns were decoupled, washed twice with 1 ml 3.5 M $HNO_3$, and Sr and the REE recovered in $H_2O$ and 2.5 M HCl, respectively. Nd was separated from the other REE using Eichrom LnSpec resin in 0.25 M HCl. Blanks were below 120, 18 and 15 pg for Sr, Nd and Pb, respectively.

Sr and Nd isotope measurements were carried out using a Thermo Triton thermal ionization mass spectrometer in static mode. Sr was loaded onto single Ta filaments in 1 ml M $H_3PO_4$, and the measured isotope ratios corrected for instrumental mass fractionation using $^{86}Sr/^{88}Sr = 0.1194$. $^{87}Rb$ interference was monitored and corrected for by measuring $^{85}Rb$, but was negligible for most samples. The NBS987 Sr standard yielded an $^{87}Sr/^{86}Sr$ value of 0.710266 ± 0.000012 ($n = 10$) over the period of analysis. Nd was loaded onto the Ta filament of a double Ta–Re filament assembly and analysed as the metal. A correction for mass fractionation was applied assuming $^{144}Nd/^{146}Nd = 0.7219$. An in-house Nd standard yielded $^{143}Nd/^{144}Nd = 0.511539 ± 0.000008$ ($n = 6$), equivalent to 0.511851 for the La Jolla Nd standard.

The Pb fraction was dissolved in 2% $HNO_3$ to obtain a Pb concentration of about 30 ng/g, and spiked with approximately 3 ng/g SRM997 Tl standard. Isotope measurements were carried out on a Thermofisher Neptune Plus multicollector plasma mass spectrometer in static mode, using a $^{205}Tl/^{203}Tl$ ratio of 2.3871 to correct measured Pb isotope ratios for instrumental mass bias. Sample measurements were bracketed by measurements of the SRM981 Pb standard, which were used for external normalization assuming the values of ref. [91]. Accuracy and reproducibility, as determined by multiple analysis of rock standards, was better than 80 ppm.

**Geochronology**. Sample preparation: The groundmass samples were prepared following the methods of ref. [92]. The 200–180 μm samples measured at Oregon State were cleaned in a series of hour-long acid baths, progressing from 1 N HCl to 6 N HCl to 1 N $HNO_3$ to 3 N $HNO_3$, followed by a final milli-Q water bath. The 355–200 μm samples measured at the VU University were cleaned in 1 N $HNO_3$ for 2 h, followed by an hour-long milli-Q water bath. Each separate was picked by hand under a binocular microscope to ensure the removal of alteration and to confirm the purity of the separate.

Oregon State University: Groundmass samples were irradiated for 6 h in the CLICIT position at the Oregon State University TRIGA reactor. Incremental heating experiments were conducted for each sample. Irradiated samples were loaded onto copper planchettes for analysis using a Thermo Scientific ARGUS-VI multicollector mass spectrometer at the OSU Argon Geochronology Laboratory following the procedure described in ref. [93]. All ages are calculated relative to Fish Canyon Tuff (FCT) sanidine with an age of 28.201 Ma[94] and using the decay constants after ref. [95].

The correction factors for neutron interference reactions at the TRIGA are $(2.64 ± 0.02) × 10^{-4}$ for $(^{36}Ar/^{37}Ar)_{Ca}$, $(6.73 ± 0.04) × 10^{-4}$ for $(^{39}Ar/^{37}Ar)_{Ca}$, $(1.21 ± 0.003) × 10^{-2}$ for $(^{38}Ar/^{39}Ar)_K$ and $(8.6 ± 0.7) × 10^{-4}$ for $(^{40}Ar/^{39}Ar)_K$. Ages were calculated using the ArArCALC v2.7.052 software of ref. [96], with errors including uncertainties on the blank corrections, irradiation constants, J-curve, collector calibrations, mass fractionation, and the decay of $^{37}Ar$ and $^{39}Ar$.

VU University: The groundmass samples were irradiated together with Fish Canyon Tuff (FCT) sanidine for 18 h at the Oregon State University TRIGA reactor in the cadmium shielded CLICIT facility (irradiation ID VU107). $^{40}Ar/$ $^{39}Ar$ analyses were performed at the geochronology laboratory of the VU University on a Helix MC noble gas mass spectrometer. Single grains of FCs were fused and multiple grain groundmass samples were step-heated with a Synrad $CO_2$ laser beam and released gas was exposed to a Lauda cooler (−70 °C)

and hot NP10 and St172 getters and analysed on the Helix MC. The five argon isotopes are measured simultaneously with $^{40}Ar$ on the H2-Faraday position with a $10^{13}$ Ω resistor amplifier, $^{39}Ar$ on the H1-Faraday with a $10^{13}$ Ω resistor amplifier, $^{38}Ar$ on the AX-CDD, $^{37}Ar$ on the L1-CDD and $^{36}Ar$ on the L2-CDD (CDD—compact discrete dynode). Gain calibration for the CDDs is done by peak jumping a $CO_2$ reference beam on all detectors in dynamic mode. All intensities are corrected relative to the L2 detector. Air pipettes are run ~every 10 h and are used for mass discrimination corrections.

All ages are calculated relative to FCT of 28.201 Ma[94] with ref. [95] decay constants. The atmospheric air value of 298.56 from ref. [97] is used. Detailed analytical procedures for the Helix MC are described in ref. [98]. The correction factors for neutron interference reactions are $(2.64 ± 0.02) × 10^{-4}$ for $(^{36}Ar/^{37}Ar)_{Ca}$, $(6.73 ± 0.04) × 10^{-4}$ for $(^{39}Ar/^{37}Ar)_{Ca}$, $(1.21 ± 0.003) × 10^{-2}$ for $(^{38}Ar/^{39}Ar)_K$ and $(8.6 ± 0.7) × 10^{-4}$ for $(^{40}Ar/^{39}Ar)_K$. All errors are quoted at the $1\sigma$ level and include all analytical errors. Data were reduced using the ArArCALC v2.7.052 software of ref. [96], with errors including uncertainties on the blank corrections, irradiation constants, J-curve, collector calibrations, mass fractionation, and the decay of $^{37}Ar$ and $^{39}Ar$.

**Data quality**. $^{40}Ar/^{39}Ar$ step-heating experiments are widely assessed using criteria as follows:

An acceptable age plateau includes at least 50% of the gas released with a mean square weighted deviation (MSWD) of approximately 1.0.
Shows an inverse isochron with a $^{40}Ar/^{36}Ar$ intercept of about 295.5 ± 2$\sigma$ (indicating equilibrium with atmospheric argon).
Has concordant plateau and inverse isochron ages within 2$\sigma$ internal error (validating the assumption of the 295.5 ratio used in determining the plateau age).

It is evident from the images of the samples used in this study that the smallest sample DL2-2 is significantly more altered compared to the other two samples. This is an obvious explanation for why mildly leached 355–200 μm groundmass did not give a plateau age. The influence of alteration on the inverse isochron $^{40}Ar/$ $^{39}Ar$ intercept values is demonstrated by how they vary for DL3-5 depending on acid leaching. Mild acid leaching gives an intercept value indicating excess argon (i.e., greater than the atmospheric value of 295.5) whereas strong acid leaching gives an atmospheric/sub-atmospheric value. Yet, both analyses give the same plateau age so validating the assumption of the 295.5 ratio used in their calculation. Another important reason that plateau ages with non-atmospheric inverse isochron intercepts are acceptable in the case of this study is that heating steps are high in $^{40}Ar(r)$ so the data points used for calculating the inverse isochrones plot close to the $^{39}Ar/^{40}Ar$ axis making it very difficult to pin any case to precisely define the $^{40}Ar/^{36}Ar$ intercept value. In summary, we consider that the plateau ages for the massive boulder sample DL3-5 most precisely defines the crystallization age of the dome samples.

## Data availability

The authors declare that all the data for the $^{40}Ar/^{39}Ar$ age determinations and geochemical analyses supporting the findings of this study are available within the paper and its supplementary information files.

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

## Acknowledgements

This project was funded through a grant by the German Federal Ministry of Education and Research (BMBF 03G0230A, 03G0231A) and AWI internal funding. We thank Captain Detlef Korte and his Crew of RV Sonne for their support and Melanie Hertel for providing the XRF analyses.

## Author contributions

K.F.K., D.P.M. and A.A.P.K. carried out the $^{40}Ar/^{39}Ar$ age determinations. M.R. carried out the geochemical analyses. W.J. conceived the project and was the PI on the SO230 expedition. J.M.O.'C. prepared the samples and wrote the paper with contributions to the discussions from W.J. and M.R.

## Competing interests

The authors declare no competing interests.
