## [Peer Review File · Nature Communications]

Reviewers' comments:

Reviewer #1 (Remarks to the Author):

O'Connor and colleagues present new geochronology and geochemistry data that provide important new insights into the dynamics and evolution of Earth's youngest hot spot province: the Africa superplume. As such, the paper is of broad interest across the full range of geosciences disciplines, and the work is timely. The study has considerable potential for a high impact paper, but I think that the authors have tried too hard to package their results into a geodynamic setting, leading to an overview that omits some new information and that ignores the ongoing controversy, as I outline below.

That being said, I think that a new submission that re-focuses the paper could be a very nice contribution to Nature Comms.

Abstract:

Sentence 2 is forced, and not correct as written. 'Channeled flow' means lateral flow of hot rock susceptible to decompression melting if it rises to shallow enough levels. So, the two parts of the sentence are equivalent.

Last sentence is also forced: there was most likely a widespread melting event associated with a spreading plume head sometime between 45-30 Ma.

The big issue is where the plume stem is located, and if there are many, or even, if the convection patterns above the mantle transition zone are more complex than in the plume models. Since the authors cannot address these geodynamical or observational questions, they could instead focus on the lateral flow of material from one or more plume stems (e.g., Chang and van der Lee, French et al., etc).

Text

There are some loose ends that need to be tightened, with the 2 most important being the channeled flow vs deep plume arguments, which are contrived or a misunderstanding, and the new data showing synchronous magmatism in the Western and Eastern rift. See below for links to specific points in the text.

20: Figure 1 starts plume-lithosphere interactions at 45 Ma, but the text says 330 Ma, which is the timing of onset of rifting associated with the separation of Arabia from Africa driven by Tethys subduction (e.g., Bellahsen et al., 2004). And, what about Africa movement since 45 Ma?

24 . 'rising beneath pre-stretched lithosphere' is confusing, in my mind. Continental lithosphere is inherently heterogeneous, specifically in terms of thickness. The cratonic roots are deep. The thinner orogenic belts adjacent to cratons are not 'pre-stretched' - they are just thinner than Archaean lithosphere. The Karroo and Palaeogene rifts are pre-existing thin zones. So, I would remove pre-thinned, since it implies a process, and instead indicate variable thickness lithosphere, and link to Fishwick and Bastow, 2011.

36: Older records indicated some southward younging in the Eastern and Western rifts, but geochron was sparse, particularly in the Western rift. New data confirm a ca. 17 Ma date for onset of magmatism in the Rungwe province, and 24 Ma in the Rukwa province (Roberts et al., 2012; Mesko et

al., 2016).

46-52: "Two models": I think the authors are confused about the geodynamics related to magma production. A rising plume, no matter where it originates, does not generate melt until P-T is above the solidus. So, thinner zones are needed to enable large volume melting. The Superplume is a critical component of 'channeled flow', and I certainly do not see 2 geodynamic models that are being tested.

A lot of new work on mantle imaging has taken place: most recent summary of Afar region is in Gallacher et al., 2017.

48-63, 94-97: How well constrained is the ocean-continent boundary along the Indian Ocean margin, and how does the shape of the lithosphere-asthenosphere boundary guide mantle flow?

The interpretation of 'transitional crust' along the Mozambique margin is based on magnetic anomaly interpretations and non-unique gravity models, correct? What kind of ambiguity does this introduce? The reader needs to know. And, given that there is 7-0 Ma magmatism, how does that influence the pattern of seafloor-spreading anomalies?

96: Very bold, unsupported statement : 'consistent with deep-seated faults associated with the tectonically active offshore EAR providing a conduit for uncontaminated low-degree melts... ' Where do deep-seated faults provide conduits for magma? We certainly don't see this in active deformation studies. Dike bring uncontaminated magmas from deep reservoirs to the shallow subsurface, and occasionally, to the surface. New faults form above the intrusions, but rarely do they flow along fault planes. Deep seated faults are not a requirement, and there are no data supporting this statement.

106-109: Incorrect. There are several papers outlining the pattern of magmatism since 45 Ma in the Ethiopian plateau region: Kieffer, B., Arndt, N., Lapierre, H., Bastien, F., Bosch, D., Pecher, A., Yirgu, G., Ayalew, D., Weis, D., Jerram, D.A. and Keller, F., 2004. Flood and shield basalts from Ethiopia: magmas from the African superswell. *Journal of Petrology*, 45(4), pp.793-834; Ebinger et al., *GSA Bull*, 2000; Wolfenden et al., 2004.

115: Rungwe is much older, and carbonatitic magmatism at 24 Ma occurred in the Rukwa area - see comments above.

131: Would tone-down and say 'is sourced from a single plume province'

144: What is the relevance of small domes? Are they specific to plume processes? Shouldn't the authors just focus on basaltic magmatism, since magma storage and fractionation will vary depending on crust and mantle lithospheric thickness and composition?

156-67: Once seafloor spreading starts, the style of magmatism is quite different. Or, are the authors making bold statements about widespread, off-axis magmatism? If yes, a lot more support is needed, and it could not be a Nature Comms short paper.

Refs:

Chang, S.J. and Van der Lee, S., 2011. Mantle plumes and associated flow beneath Arabia and East Africa. *Earth and Planetary Science Letters*, 302(3-4), pp.448-454.

Cindy Ebinger

Mesko, G. T., C. Class, M. D. Maqway, N. Boniface, S. Many, and S. R. Hemming. "The Timing of Early Magmatism and Extension in the Southern East African Rift: Tracking Geochemical Source Variability with $^{40}\text{Ar}/^{39}\text{Ar}$ Geochronology at the Rungwe Volcanic Province, SW Tanzania." In AGU Fall Meeting Abstracts. 2014.

Roberts, E.M., Stevens, N.J., O'Connor, P.M., Dirks, P.H.G.M., Gottfried, M.D., Clyde, W.C., Armstrong, R.A., Kemp, A.I.S. and Hemming, S., 2012. Initiation of the western branch of the East African Rift coeval with the eastern branch. *Nature Geoscience*, 5(4), p.289.

Sembroni, A., Faccenna, C., Becker, T.W., Molin, P. and Abebe, B., 2016. Long-term, deep-mantle support of the Ethiopia-Yemen Plateau. *Tectonics*, 35(2), pp.469-488.

Reviewer #2 (Remarks to the Author):

Review of

"Evidence for channeled flow of Afar-like plume mantle over 5,000 km" by John O'Connor et al.,

General remarks:

The manuscript by O'Connor et al. presents new Ar-Ar dates and chemical and isotopic characterization of recently discovered basanites from submarine volcanic dome along the seaward extension of the East African rift system in northern Mozambique ridge. Their motivation is to test the influence of the Afar plume at the far south end of the East African Rift.

Although relatively small, this is a nice dataset, that appears to be of high quality, and therefore should be published. This is also mostly a fine manuscript, well written, and for the most part clearly presented and thus easy to follow and one which has the potential to appeal to a rather wide group of earth scientists studying the geodynamic evolution of continental rift systems as well as the nature of mantle plumes.

I think the paper can be published but with some changes and further consideration/clarifications by the authors. Particularly important in this regard is a more in-depth and detailed comparison with available datasets from some key regions (not just the Red Sea and the Gulf of Aden) of the EARS in Figures 3 and 4. Below I discuss some points and make a few suggestions about what I think would help improve the paper.

General comments:

- There is good evidence from radiogenic isotopes and noble gases that Afar plume compositions can be traced to a common plume source, i.e. the African superplume, affecting the entire EARS. This is briefly discussed in the manuscript, but as the upper mantle seismic anomalies are most pronounced under Ethiopia, O'Connor prefer to use the use the term Afar plume as the ultimate plume source. Why not adopt the term African superplume instead? This needs better justification in the manuscript.
- The reason I ask the author to consider this is that when consider recent seismic tomographic models of the region, for example one discussing P-wave velocity variations in the Afro-Arabian mantle by Hansen et al. (EPSL, 2012), it becomes clear that the low-velocity anomaly beneath the entire EARS are best explained by the African superplume model, in which the "anomalous upper mantle structure is a continuation of a large, thermo-chemical upwelling in the lower mantle beneath southern Africa". In other words, the plume rises from the lower mantle beneath southern Africa and

appears to form "a laterally continuous, low-velocity region in the upper mantle beneath all of eastern Africa and western Arabia, extending to depths of ~500–700 km".

- If these seismic tomographic models of the region are indeed valid (they are also discussed by the authors in the manuscript), I find it a bit confusing that the authors argue for a channeled flow of the an Afar-like plume for over 5,000 km? According to these tomographic models, the upwelling loci should be much further south of the Afar region and therefore plume material has travelled somewhat shorter distances (unless the authors are arguing that plume material travels first to Afar and then some 5,000 km back to the Indian Ocean?) In any case, this needs to be clarified as the seismic tomographic models are cited but seem to contradict what the authors are arguing for.
- My point is, although I do agree with the authors that there is evidence for some channeling of plume material in the region, the authors seem to have taken an extreme point of view when arguing for it. At least, this does not come out very clear to me in the manuscript.

Below are some major additional details that need to be considered.

Detailed comments:

Lines 6-7: why not say "Afar mantle plume"? This phrase is used in line 12, for example. Please be consistent throughout the manuscript. I would rather use African superplume.

Line 37-38: EAR, already defined in line 16

Line 41: define LLSVP

Line 46, add comma after mechanics

Line 52-53: "both models"? What models? This has not been introduced very well up to this point. Please provide a better background to both model. As I read it, only the plume support model has been introduced. In addition, recent work by Hilton et al. (2011) and Castillo et al (2014) did exactly this; see if Afar-like compositions are evident in the southern end of the EARS and should be cited here.

Lines 54-55: But there might still be weak indications of Afar-like compositions in the southern end, but these are likely to be more difficult to identify from the often overwhelming lithospheric signatures. Please consider rephrasing to emphasize that a plume signature might be more difficult to spot in the southern end due to enhanced lithospheric thickness. In addition, why does the plume signature need to be derived from a channeled Afar plume? Why not just an upwelling African superplume as implied by some seismic tomographic models?

Line 90: add "generally believed" or something similar before "derived"

Lines 91-95: See also comments to Figure 3. This model needs better justification as well as explanations. These new data need to be put into a much broader region perspective emphasizing regions of the EARS which show the same characteristics as the new dome samples do. Why is this part of Results but not Discussions? Also, add the word "likely" before resulting in line 94.

Lines 99-103: Similar comment as above. Much better explanations are needed. A specific section, discussing argument against contamination, appears competently missing. It needs to be added.

Lines 106-117: A comparison with chemical or more importantly isotopic data from these volcanics is badly missing. Without such comparison, these discussions seem very speculative and without focus.

Lines 130-132: I'm confused? Has this been shown already? Isn't this what you want to test? Given what we know from seismic tomography and isotope geochemistry of the region, I'm not convinced that you need channeling for over 5,000 km. Might just as well be upwelling of the African Superplume interacting with lithosphere of different thickness.

Lines 136-138: This is not cited correctly. High $^3\text{He}/^4\text{He}$ ratios are for example evident in MER basalts and geothermal fluids, so in lithospheric contamination is less of a problem there relative to other regions of the EARS. Contamination from lithospheric signatures is much more pronounced in most other part of EARS (e.g., Furman; Rogers; Castillo; Halldórsson etc.) and this needs to be emphasized much better by comparison with available datasets.

Lines 144-167: Although it is important the plume material sampled at the opposite ends of the EARS samples similar mantle compositions, the comparison with the Red Sea and the Gulf of Aden is overemphasized. Equally important is a comparison with on-land regions that show clearer plume signature, for example Afar, MER and Rungwe.

Lines 168-186: This discussion is bit too weak at present. If this is the case, what is the actual plume contribution in these samples? Setup mixing lines on Figure 4 to emphasize and test this point. In fact, same applies to all other samples plotted. At present, this model is not fully justified, with the comparison with the new data from the MR being too loosely focused. What do the authors want the readers to take from this?

Lines 200-203: I'm confused; which model do the authors favor?

Lines: 204-205: Does the Hansen et al paper suggest that the plume material erupted in Ethiopia has lost connection to the deep mantle? I don't think so. Their models suggest that the plume rises from the lower mantle beneath southern Africa and forms "a laterally continuous, low-velocity region in the upper mantle beneath all of eastern Africa and western Arabia, extending to depths of ~500–700 km". How is this compatible with the model presented in this manuscript? This needs better explanation.

Figure 1. I find this figure somewhat confusing and think that it needs to be improved.

1. Why is 1b not given the supplement. It adds very little to the story here. In fact, it adds a lot of confusion when reading 1a.
2. Why is extent of plume material between present and 45 Ma important here? This derives from A MODEL by Ebinger and Sleep and do not understand why it is important for this figure. Please consider focusing on observational data but not models which are of little value for this figure. A simplified geological map, showing distribution of young volcanics along the EARS, would be much more beneficial. Alternatively, the distribution of seismic anomalies in uppermost 500-700 km beneath the region (e.g., Hansen et al. 2012) might be very informative.
3. The 2800 km seismic anomalies (there might be updated with more recent datasets) indicate upwelling from southern Africa. Again, how does this help the argument favoring a channeled plume flow from Afar?
4. A scale bar is missing.

Figure 2. Maybe the Ar-Ar age spectra can be moved to the supplement to save space? They add very little to the manuscript.

Figure 3. This model needs better justification as well as explanations. The title should simply say "Trace elemental characteristics". What does the red star indicate? Why not plot all the dome samples? Why call the dome samples "OIB"? These new data need to be put into a much broader

regional perspective emphasizing regions of the EARS which show the same characteristics as the new dome samples.

Figure 4. The title should simply say "Radiogenic isotope characteristics". No need to repeat the symbol legend from previous figure. Symbols in both 3 and 4 need to be improved, emphasizing the dome samples: they are difficult to spot as it is now. Why is it necessary to separate the different volcanic from the Afar/Red Sea/Aden region? These might just as well be shown as one or possibly two, symbols. In addition, these new data need to be put into a much broader regional perspective. For example, a key dataset from the Rungwe volcanic region that shows a clear plume signature despite located at the southernmost extreme of the EARS (See Castillo et al. 2014), is not plotted. Same applied to data from Ethiopia.

Geochemistry part: Please list all standards, both internal and external, measured along with samples and the values obtained. This includes any USGS standards, such as BHVO and BCR etc.

Reviewer #3 (Remarks to the Author):

O'Connor et. al present new geochemical analyses that constrain the age and isotopic signature of rocks from across the East African Rift System, including a newly discovered volcanic dome on the Mozambique Ridge. The results suggest that the volcanism is young ($\sim 7\text{Ma}$), is free of lithospheric contamination and has the same isotopic signature of other volcanics from across East Africa. They use the results to suggest that the volcanism has a common plume source (described as "Afar-like") and that material from this plume has undergone channeled flow due to lithospheric thickness variations along more than 5,000km.

While the study present interesting new results which seem robust and extend the fingerprinting of the Afar plume further south than has been previously seen, it has been already shown that almost all of the East African Rift System does contain an Afar-like geochemical signature. In this sense, the study can be considered incremental in nature. The geodynamical implications are also sufficiently vague that it adds little to the increasingly complex view that is developing regarding the geometry of upwelling beneath east Africa. Due to these facts, it could be that the study is not ideally suited for publication in Nature Communications. I have documented my thoughts below and hope they are constructive to further iterations of the paper.

KEY COMMENTS

i) The constraints from this paper suggest a single Afar plume source across the entire region, Other results from isotope geochemistry cited in the paper also confirm that there is a common, deep-seated mantle plume source for volcanism across the EARS (Halldorsson et. al, 2014). It is not clear to me that the new observations in this paper constrain anything different to this (albeit from a new locality and a different isotope system), hence my statement previously about the study being incremental in nature.

ii) There is also still considerable debate about the number of upwellings beneath East Africa and their interconnectivity at depth. It is becoming increasingly clear that upwelling is occurring beneath many different parts of the EARS (e.g. Civiero et. al, 2015; Civiero et. al 2016; Nyblade, 2011, Chang and Van der Lee, 2011). It seems, for instance, that the Kenyan and Ethiopian domes represent two separate upwellings from the same lower mantle source region (likely the African Superplume). Again,

the conclusions of the paper suggest that the new results can be explained by “channeling of one or more discrete plumes under the Ethiopia-Afar region, or tilting of a plume under southern Africa”. This means that this study adds little to our ongoing understanding of the connectivity between upper and lower mantle beneath East Africa.

GENERAL COMMENTS

i) Line 40: The authors should be careful with the statement made here. Tomographic models show that low seismic velocities are present to the core-mantle boundary beneath East Africa. It is only an interpretation that this is a plume, and there are a number of scientists globally that would disagree with this (I’m not one of them for the record).

ii) Figure 1: The locations of the LLSVP at the CMB don’t really add anything to this plot and simply add further lines to try and understand. While there are arguments that plumes rise from the edge of these, it isn’t really relevant to this study. I would get rid of these in order to simplify the plot for the reader.

iii) I really think that the paper would benefit from a schematic illustration of the conclusions, it took a couple of reads to digest exactly what the paper was suggesting. A schematic diagram would clear this up.

REFERENCES

Bagley, B. and Nyblade, A.A., 2013. Seismic anisotropy in eastern Africa, mantle flow, and the African superplume. *Geophysical Research Letters*, 40(8), pp.1500-1505.

Chang, S.J. and Van der Lee, S., 2011. Mantle plumes and associated flow beneath Arabia and East Africa. *Earth and Planetary Science Letters*, 302(3-4), pp.448-454.

Civiero, C., Hammond, J.O., Goes, S., Fishwick, S., Ahmed, A., Ayele, A., Doubre, C., Goitom, B., Keir, D., Kendall, J. and Leroy, S., 2015. Multiple mantle upwellings in the transition zone beneath the northern East-African Rift system from relative P-wave travel-time tomography. *Geochemistry, Geophysics, Geosystems*, 16(9), pp.2949-2968.

Civiero, C., Goes, S., Hammond, J.O., Fishwick, S., Ahmed, A., Ayele, A., Doubre, C., Goitom, B., Keir, D., Kendall, J.M. and Leroy, S., 2016. Small-scale thermal upwellings under the northern East African Rift from S travel time tomography. *Journal of Geophysical Research: Solid Earth*, 121(10), pp.7395-7408.

Halldórsson, S.A., Hilton, D.R., Scarsi, P., Abebe, T. and Hopp, J., 2014. A common mantle plume source beneath the entire East African Rift System revealed by coupled helium-neon systematics. *Geophysical Research Letters*, 41(7), pp.2304-2311.

Nyblade, A.A., 2011. The upper-mantle low-velocity anomaly beneath Ethiopia, Kenya, and Tanzania: Constraints on the origin of the African superswell in eastern Africa and plate versus plume models of mantle dynamics. *Geological Society of America Special Papers*, 478, pp.37-50.

We appreciate very much the constructive and insightful comments by the three expert referees, which have been very useful in revising our manuscript.

Reviewer #1 (Remarks to the Author):

Reviewers' comments:

Reviewer #1 (Remarks to the Author):

O'Connor and colleagues present new geochronology and geochemistry data that provide important new insights into the dynamics and evolution of Earth's youngest hot spot province: the Africa superplume. As such, the paper is of broad interest across the full range of geosciences disciplines, and the work is timely.

The study has considerable potential for a high impact paper, but I think that the authors have tried too hard to package their results into a geodynamic setting, leading to an overview that omits some new information and that ignores the ongoing controversy, as I outline below.

That being said, I think that a new submission that re-focusses the paper could be a very nice contribution to Nature Comms.

Abstract:

Sentence 2 is forced, and not correct as written. 'Channeled flow' means lateral flow of hot rock susceptible to decompression melting if it rises to shallow enough levels. So, the two parts of the sentence are equivalent.

Sentence has been deleted.

Last sentence is also forced: there was most likely a widespread melting event associated with a spreading plume head sometime between 45-30 Ma.

Sentence has been deleted.

The big issue is where the plume stem is located, and if there are many, or even, if the convection patterns above the mantle transition zone are more complex than in the plume models.

Agreed. This point is now in the revised manuscript.

Since the authors cannot address these geodynamical or observational questions, they could instead focus on the lateral flow of material from one or more plume stems (e.g., Chang and van der Lee, French et al., etc).

Agreed. We now focus on the lateral flow of plume material.

Text

There are some loose ends that need to be tightened, with the 2 most important being the channeled flow vs deep plume arguments, which are contrived or a misunderstanding, and the new data showing synchronous magmatism in the Western and Eastern rift. See below for links to specific points in the text.

20: Figure 1 starts plume-lithosphere interactions at 45 Ma, but the text says 30 Ma, which is the timing of onset of rifting associated with the separation of Arabia from Africa driven by Tethys subduction (e.g., Bellahsen et al., 2004).

Corrected.

And, what about Africa movement since 45 Ma?

About 1000 km based on 23 km/Ma inferred from the Tristan-Gough track. But we don't see an argument for including this point. But we are open to a suggestion.

24. 'rising beneath pre-stretched lithosphere' is confusing, in my mind. Continental lithosphere is inherently heterogeneous, specifically in terms of thickness. The cratonic roots are deep. The thinner orogenic belts adjacent to cratons are not 'pre-stretched' - they are just thinner than Archaean lithosphere. The Karroo and Palaeogene rifts are pre-existing thin zones. So, I would remove pre-thinned, since it implies a process, and instead indicate variable thickness lithosphere, and link to Fishwick and Bastow, 2011.

Agreed. We now mention this point and reference.

36: Older records indicated some southward younging in the Eastern and Western rifts, but geochron was sparse, particularly in the Western rift. New data confirm a ca. 17 Ma date for onset of magmatism in the Rungwe province, and 24 Ma in the Rukwa province (Roberts et al., 2012; Mesko et al., 2016).

We now mention these ages and references.

46-52: "Two models": I think the authors are confused about the geodynamics related to magma production.

We no longer mention testing "Two models".

A rising plume, no matter where it originates, does not generate melt until P-T is above the solidus. So, thinner zones are needed to enable large volume melting. The Superplume is a critical component of 'channeled flow', and I certainly do not see 2 geodynamic models that are being tested.

We agree that superplume encompasses the notion of 'channeled flow'. We no longer propose testing 2 geodynamic models and don't mention 'channeled flow' as it is inherent to a superplume.

A lot of new work on mantle imaging has taken place: most recent summary of Afar region is in Gallacher et al., 2017.

Done.

48-63, 94-97: How well constrained is the ocean-continent boundary along the Indian Ocean margin, We draw on a 2019 synthesis paper by Mueller & Jokat to address this question. We include a figure from this paper in our revised Figure 2, which shows two possible COB locations in the vicinity of our dredge-sampling site.

and how does the shape of the lithosphere-asthenosphere boundary guide mantle flow?

Done. Please see text.

The interpretation of 'transitional crust' along the Mozambique margin is based on magnetic anomaly interpretations and non-unique gravity models, correct?

Yes

What kind of ambiguity does this introduce?

We draw again on the synthesis paper by Mueller & Jokat (2019) reporting that potential field data and magnetic lineation evidence can be used to show that “the entire Natal Valley and Mozambique Ridge is floored by oceanic crust emplaced between 157.1–144 Ma and 144–131.7 Ma, respectively, with an anomalous thick oceanic layer 3 and/or significant magmatic underplate.”

And, given that there is 7-0 Ma magmatism, how does that influence the pattern of seafloor-spreading anomalies?

Seafloor spreading was developed between ~157–144 Ma so seemingly not much.

96: Very bold, unsupported statement : 'consistent with deep-seated faults associated with the tectonically active offshore EAR providing a conduit for uncontaminated low-degree melts... ' Where do deep-seated faults provide conduits for magma? We certainly don't see this in active deformation studies. Dike bring uncontaminated magmas from deep reservoirs to the shallow subsurface, and occasionally, to the surface. New faults form above the intrusions, but rarely do they flow along fault planes. Deep seated faults are not a requirement, and there are no data supporting this statement.

We now draw on a 2018 paper by Deville et al, to make the point that there is a correlation between tectonic activity associated with the EARS and off-shore low volume seafloor volcanism since at least Late Miocene times.

106-109: Incorrect. There are several papers outlining the pattern of magmatism since 45 Ma in the Ethiopian plateau region:

--Kieffer, B., Arndt, N., Lapierre, H., Bastien, F., Bosch, D., Pecher, A., Yirgu, G., Ayalew, D., Weis, D., Jerram, D.A. and Keller, F., 2004. Flood and shield basalts from Ethiopia: magmas from the African superswell. *Journal of Petrology*, 45(4), pp.793-834;

--Ebinger et al., *GSA Bull*, 2000;

--Wolfenden et al., 2004.

115: Rungwe is much older, and carbonatitic magmatism at 24 Ma occurred in the Rukwa area - see comments above.

These references are now included.

131: Would tone-down and say 'is sourced from a single plume province'

Done

144: What is the relevance of small domes? Are they specific to plume processes? Shouldn't the authors just focus on basaltic magmatism, since magma storage and fractionation will vary depending on crust and mantle lithospheric thickness and composition?

Done. We no longer focus on small domes. Our argument is that the small domes/cones and low volume seafloor volcanism reflect high mean pressure and low mean extent of melting that samples the uncontaminated/undiluted superplume isotopic signature.

156-67: Once seafloor spreading starts, the style of magmatism is quite different. Or, are the authors making bold statements about widespread, off-axis magmatism? If yes, a lot more support is needed, and it could not be a Nature Comms short paper.

We have omitted this section.

Refs:

Chang, S.J. and Van der Lee, S., 2011. Mantle plumes and associated flow beneath Arabia and East Africa. *Earth and Planetary Science Letters*, 302(3-4), pp.448-454.

Cindy Ebinger

?

Mesko, G. T., C. Class, M. D. Maqway, N. Boniface, S. Manya, and S. R. Hemming. "The Timing of Early Magmatism and Extension in the Southern East African Rift: Tracking Geochemical Source Variability with $^{40}\text{Ar}/^{39}\text{Ar}$ Geochronology at the Rungwe Volcanic Province, SW Tanzania." In AGU Fall Meeting Abstracts. 2014.

Roberts, E.M., Stevens, N.J., O'Connor, P.M., Dirks, P.H.G.M., Gottfried, M.D., Clyde, W.C., Armstrong, R.A., Kemp, A.I.S. and Hemming, S., 2012. Initiation of the western branch of the East African Rift coeval with the eastern branch. *Nature Geoscience*, 5(4), p.289.

Sembroni, A., Faccenna, C., Becker, T.W., Molin, P. and Abebe, B., 2016. Long-term, deep-mantle support of the Ethiopia-Yemen Plateau. *Tectonics*, 35(2), pp.469-488.

References are now included.

Reviewer #2 (Remarks to the Author):

Review of

"Evidence for channeled flow of Afar-like plume mantle over 5,000 km" by John O'Connor et al.,

General remarks:

The manuscript by O'Connor et al. presents new Ar-Ar dates and chemical and isotopic characterization of recently discovered basanites from submarine volcanic dome along the seaward extension of the East African rift system in northern Mozambique ridge. Their motivation is to test the influence of the Afar plume at the far south end of the East African Rift.

Although relatively small, this is a nice dataset, that appears to be of high quality, and therefore should be published. This is also mostly a fine manuscript, well written, and for the most part clearly presented and thus easy to follow and one which has the potential to appeal to a rather wide group of earth scientists studying the geodynamic evolution of continental rift systems as well as the nature of mantle plumes.

I think the paper can be published but with some changes and further consideration/clarifications by the authors. Particularly important in this regard is a more in-depth and detailed comparison with available datasets from some key regions (not just the Red Sea and the Gulf of Aden) of the EARS in Figures 3 and 4.

Agreed. Please see revised figures and the main text.

Below I discuss some points and make a few suggestions about what I think would help improve the paper.

General comments:

- There is good evidence from radiogenic isotopes and noble gases that Afar plume compositions can be traced to a common plume source, i.e. the African superplume, affecting the entire EARS.

This is briefly discussed in the manuscript, but as the upper mantle seismic anomalies are most pronounced under Ethiopia, O'Connor prefer to use the use the term Afar plume as the ultimate plume source. Why not adopt the term African superplume instead? This needs better justification in the manuscript.

Agreed. We have adopted the term African superplume instead.

- The reason I ask the author to consider this is that when consider recent seismic tomographic models of the region, for example one discussing P-wave velocity variations in the Afro-Arabian mantle by Hansen et al. (EPSL, 2012), it becomes clear that the low-velocity anomaly beneath the entire EARS are best explained by the African superplume model, in which the “anomalous upper mantle structure is a continuation of a large, thermo-chemical upwelling in the lower mantle beneath southern Africa”. In other words, the plume rises from the lower mantle beneath southern Africa and appears to forms “a laterally continuous, low-velocity region in the upper mantle beneath all of eastern Africa and western Arabia, extending to depths of ~500–700 km”.

Agreed. We now relate our finding with the low-velocity zone

- If these seismic tomographic models of the region are indeed valid (they are also discussed by the authors in the manuscript), I find it a bit confusing that the authors argue for a channeled flow of the an Afar-like plume for over 5,000 km?

Agreed. Please see text.

According to these tomographic models, the upwelling loci should be much further south of the Afar region and therefore plume material has travelled somewhat shorter distances (unless the authors are arguing that plume material travels first to Afar and then some 5,000 km back to the Indian Ocean?) In any case, this needs to be clarified as the seismic tomographic models are cited but seem to contradict what the authors are arguing for.

Agreed. Please see text.

- My point is, although I do agree with the authors that there is evidence for some channeling of plume material in the region, the authors seem to have taken an extreme point of view when arguing for it. At least, this does not come out very clear to me in the manuscript.

Agreed. Please see text.

Below are some major additional details that need to be considered.

Detailed comments:

Lines 6-7: why not say “Afar mantle plume”? This phrase is used in line 12, for example. Please be consistent throughout the manuscript. I would rather use African superplume.

Agreed.

Line 37-38: EAR, already defined in line 16

Fixed.

Line 41: define LLSVP

Done.

Line 46, add comma after mechanics

Done.

Line 52-53: “both models”? What models? This has not been introduced very well up to this point. Please provide a better background to both model. As I read it, only the plume support model has been introduced.
Agreed. We no longer refer to both models.

In addition, recent work by Hilton et al. (2011) and Castillo et al (2014) did exactly this; see if Afar-like compositions are evident in the southern end of the EARS and should be cited here.
Done. Please see text.

Lines 54-55: But there might still be weak indications of Afar-like compositions in the southern end, but these are likely to be more difficult to identify from the often overwhelming lithospheric signatures. Please considering rephrasing to emphasize that a plume signature might be more difficult to spot in the southern end due to enhanced lithospheric thickness.
We have clarified in the revised text that the plume signature is easier to spot due to enhanced lithospheric thickness.

In addition, why does the plume signature need to be derived from a channeled Afar plume? Why not just an upwelling African superplume as implied by some seismic tomographic models?
Agreed. Please see text.

Line 90: add “generally believed” or something similar before “derived”
Done.

Lines 91-95: See also comments to Figure 3. This model needs better justification as well as explanations. These new data need to be put into a much broader region perspective emphasizing regions of the EARS which show the same characteristics as the new dome samples do.
Done.

Why is this part of Results but not Discussions? Also, add the word “likely” before resulting in line 94.
Done.

Lines 99-103: Similar comment as above. Much better explanations are needed. A specific section, discussing argument against contamination, appears competently missing. It needs to be added.
Done.

Lines 106-117: A comparison with chemical or more importantly isotopic data from these volcanics is badly missing. Without such comparison, these discussions seem very speculative and without focus.
Done.

Lines 130-132: I’m confused? Has this been shown already? Isn’t this what you want to test? Given what we know from seismic tomography and isotope geochemistry of the region, I’m not convinced that you need channeling for over 5,000 km. Might just as well be upwelling of the African Superplume interacting with lithosphere of different thickness.
We now refer to the African Superplume interacting with lithosphere of different thickness rather than a channeling plume for over 5,000 km

Lines 136-138: This is not cited correctly. High $^3\text{He}/^4\text{He}$ ratios are for example evident in MER basalts and geothermal fluids, so in lithospheric contamination is less of a problem there relative to other regions of the EARS. Contamination from lithospheric signatures is much more pronounced in most other part of

EARS (e.g., Furman; Rogers; Castillo; Halldórsson etc.) and this needs to be emphasized much better by comparison with available datasets.

Agreed. Please see text.

Lines 144-167: Although it is important the plume material sampled at the opposite ends of the EARS samples similar mantle compositions, the comparison with the Red Sea and the Gulf of Aden is overemphasized. Equally important is a comparison with on-land regions that show clearer plume signature, for example Afar, MER and Rungwe.

Done

Lines 168-186: This discussion is bit too weak at present. If this is the case, what is the actual plume contribution in these samples? Setup mixing lines on Figure 4 to emphasize and test this point. In fact, same applies to all other samples plotted. At present, this model is not fully justified, with the comparison with the new data from the MR being too loosely focused.

What do the authors want the readers to take from this?

Please see the revised text for a (hopefully) much stronger discussion. We have endeavored to justify our model and to clarify our take-home message. Our thinking is that using mixing lines to estimate actual plume contribution to explain our revised model would confuse our message too much.

Lines 200-203: I'm confused; which model do the authors favor?

This should now be clear to readers.

Lines: 204-205: Does the Hansen et al paper suggest that the plume material erupted in Ethiopia has lost connection to the deep mantle? I don't think so. Their models suggest that the plume rises from the lower mantle beneath southern Africa and forms "a laterally continuous, low-velocity region in the upper mantle beneath all of eastern Africa and western Arabia, extending to depths of ~500–700 km".

Agreed. Please see text.

How is this compatible with the model presented in this manuscript? This needs better explanation.

Done.

Figure 1. I find this figure somewhat confusing and think that it needs to be improved.

1. Why is 1b not given the supplement. It adds very little to the story here. In fact, it adds a lot of confusion when reading 1a.

Done.

2. Why is extent of plume material between present and 45 Ma important here? This derives from A MODEL by Ebinger and Sleep and do not understand why it is important for this figure. Please consider focusing on observational data but not models which are of little values for this figure.

Done.

A simplified geological map, showing distribution of young volcanics along the EARS, would be much more beneficial. Alternatively, the distribution of seismic anomalies in uppermost 500-700 km beneath the region (e.g., Hansen et al. 2012) might be very informative.

Young volcanics are now shown in Figure 1 and distribution of seismic anomalies in uppermost 500-700 km beneath the region (i.e., Hansen et al. 2012) is show in Figure 7.

3. The 2800 km seismic anomalies (there might be updated with more recent datasets) indicate upwelling from southern Africa. Again, how does this help the argument favoring a channeled plume flow from Afar?

Removed.

4. A scale bar is missing.

Done.

Figure 2. Maybe the Ar-Ar age spectra can be moved to the supplement to save space? They add very little to the manuscript.

Good suggestion. But since we have sufficient available space we have not moved them to the supplement, at least for now.

Figure 3. This model needs better justification as well as explanations. The title should simply say “Trace elemental characteristics”. What does the red star indicate? Why not plot all the dome samples? Why call the dome samples “OIB”? These new data need to be put into a much broader regional perspective emphasizing regions of the EARS which show the same characteristics as the new dome samples.

Done. Please see new Figure 4.

Figure 4. The title should simply say “Radiogenic isotope characteristics”. No need to repeat the symbol legend from previous figure. Symbols in both 3 and 4 need to be improved, emphasizing the dome samples: they are difficult to spot as it is now. Why is it necessary to separate the different volcanic from the Afar/Red Sea/Aden region? These might just as well be shown as one or possibly two, symbols. In addition, these new data need to be put into a much broader regional perspective. For example, a key dataset from the Rungwe volcanic region that shows a clear plume signature despite located at the southernmost extreme of the EARS (See Castillo et al. 2014), is not plotted. Same applied to data from Ethiopia.

Done. Please see new Figure 5.

Geochemistry part: Please list all standards, both internal and external, measured along with samples and the values obtained. This includes any USGS standards, such as BHVO and BCR etc.

Done.

Reviewer #3 (Remarks to the Author):

O’Connor et. al present new geochemical analyses that constrain the age and isotopic signature of rocks from across the East African Rift System, including a newly discovered volcanic dome on the Mozambique Ridge.

The results suggest that the volcanism is young (~7Ma), is free of lithospheric contamination and has the same isotopic signature of other volcanics from across East Africa. They use the results to suggest that the volcanism has a common plume source (described as “Afar-like”) and that material from this plume has undergone channeled flow due to lithospheric thickness variations along more than 5,000km.

While the study present interesting new results which seem robust and extend the fingerprinting of the Afar plume further south than has been previously seen, it has been already shown that almost all of the East African Rift System does contain an Afar-like geochemical signature.

In this sense, the study can be considered incremental in nature.

We double the known extent of the geochemically traced superplume under the EARS.

The geodynamical implications are also sufficiently vague that it adds little to the increasingly complex view that is developing regarding the geometry of upwelling beneath east Africa.

Due to these facts, it could be that the study is not ideally suited for publication in Nature Communications.

We show that the implications are not vague, rather point to the African superplume.

I have documented my thoughts below and hope they are constructive to further iterations of the paper.

KEY COMMENTS

i) The constraints from this paper suggest a single Afar plume source across the entire region, Other results from isotope geochemistry cited in the paper also confirm that there is a common, deep-seated mantle plume source for volcanism across the EARS (Halldorsson et. al, 2014). It is not clear to me that the new observations in this paper constrain anything different to this (albeit from a new locality and a different isotope system), hence my statement previously about the study being incremental in nature.

As mentioned already, we double the known extent of the geochemically trace superplume under the EARS.

ii) There is also still considerable debate about the number of upwellings beneath East Africa and their interconnectivity at depth. It is becoming increasingly clear that upwelling is occurring beneath many different parts of the EARS (e.g. Civiero et. al, 2015; Civiero et. al 2016; Nyblade, 2011, Chang and Van der Lee, 2011).

Agreed. This is discussed in our revised text.

It seems, for instance, that the Kenyan and Ethiopian domes represent two separate upwellings from the same lower mantle source region (likely the African Superplume).

Again, the conclusions of the paper suggest that the new results can be explained by “channeling of one or more discrete plumes under the Ethiopia-Afar region, or tilting of a plume under southern Africa”. This means that this study adds little to our ongoing understanding of the connectivity between upper and lower mantle beneath East Africa.

We related the magma-poor southern half of the EARS to the tilting of the African superplume.

GENERAL COMMENTS

i) Line 40: The authors should be careful with the statement made here. Tomographic models show that low seismic velocities are present to the core-mantle boundary beneath East Africa. It is only an interpretation that this is a plume, and there are a number of scientists globally that would disagree with this (I'm not one of them for the record).

Agreed.

ii) Figure 1: The locations of the LLSVP at the CMB don't really add anything to this plot and simply add further lines to try and understand. While there are arguments that plumes rise from the edge of these, it isn't really relevant to this study. I would get rid of these in order to simplify the plot for the reader.

Agreed. We have removed the LLSVP etc.

iii) I really think that the paper would benefit from a schematic illustration of the conclusions, it took a couple of reads to digest exactly what the paper was suggesting. A schematic diagram would clear this up.

Good suggestion, thanks. Please see Figure 7

REFERENCES

Bagley, B. and Nyblade, A.A., 2013. Seismic anisotropy in eastern Africa, mantle flow, and the African superplume. *Geophysical Research Letters*, 40(8), pp.1500-1505.

Chang, S.J. and Van der Lee, S., 2011. Mantle plumes and associated flow beneath Arabia and East Africa. *Earth and Planetary Science Letters*, 302(3-4), pp.448-454.

Civiero, C., Hammond, J.O., Goes, S., Fishwick, S., Ahmed, A., Ayele, A., Doubre, C., Goitom, B., Keir, D., Kendall, J. and Leroy, S., 2015. Multiple mantle upwellings

in the transition zone beneath the northern East-African Rift system from relative P-wave travel-time tomography. *Geochemistry, Geophysics, Geosystems*, 16(9), pp.2949-2968.

Civiero, C., Goes, S., Hammond, J.O., Fishwick, S., Ahmed, A., Ayele, A., Doubre, C., Goitom, B., Keir, D., Kendall, J.M. and Leroy, S., 2016. Small-scale thermal upwellings under the northern East African Rift from S travel time tomography. *Journal of Geophysical Research: Solid Earth*, 121(10), pp.7395-7408.

Halldórsson, S.A., Hilton, D.R., Scarsi, P., Abebe, T. and Hopp, J., 2014. A common mantle plume source beneath the entire East African Rift System revealed by coupled helium-neon systematics. *Geophysical Research Letters*, 41(7), pp.2304-2311.

Nyblade, A.A., 2011. The upper-mantle low-velocity anomaly beneath Ethiopia, Kenya, and Tanzania: Constraints on the origin of the African superswell in eastern Africa and plate versus plume models of mantle dynamics. *Geological Society of America Special Papers*, 478, pp.37-50.

Above are all cited in the revised manuscript.

Reviewers' comments:

Reviewer #2 (Remarks to the Author):

After carefully reading the revised manuscript and the answers given by the authors to my questions, I cannot see any reason why the article cannot be published. The authors have done a lot of work improving the manuscript and addressing the concerns of all referees in a convincing fashion. As such, this version is a great improvement over the previous, and I recommend publication.

However, here are few points that I strongly advice the author to consider:

- The revised title is very similar to one used by Halldorsson et al (GRL 2014). I wonder if the author can be more creative here? Maybe the southernmost extreme of the EARS is relevant to bring up in the title? Or something regarding increased extent of superplume influences under the EARS. Their revised title reads as if this contribution is adding nothing to the work done by Castillo et al and Halldorsson et al, which is not true. Please consider to revise the title. This may potentially improve the future impact of this manuscript.
- The authors state that Young volcanics are now shown in Figure 1 a; Im sorry, but I do not see this on Figure 1 and there is no Figure 1a? What am I missing here?
- I am sorry, but I still think that the Ar-Ar age spectra should be moved to the supplement? These are analytical details which add little to the main message of this manuscript. In addition, they complicate the presentation of Fig. 2. The focus there should be on the geological characteristics of the region but not the Ar-Ar- age spectra.
- Figure 6b is confusing. On Figure 1, please indicate what area 6b covers.
- However, I remain skeptical regarding the importance of Fig. 6 for this manuscript. In my opinion, it does not add much to the manuscript and appears out of place. Why not place it in the supplement? The concept of a seaward extension of the EARS has already been introduced in Fig. 1 and 2. Why repeat this in Fig. 6? This is not clear to me.
- In my opinion, Figure 7 greatly improves the manuscript but Figures 7b and 7c appear to be taken directly from Hansen et al. (EPSL 2012). Not entirely sure, but wonder if Nature Com. approves this? Why not just refer to the work of Hansen et al. (2012) or introduce a modified version of their figure in Fig. 7c?

Reviewer #4 (Remarks to the Author):

Review of "Evidence for a common (super)plume in the upper mantle under the entire East African Rift System" by O'Connor et al., (2019) for publication in Nature Communications

This is an interesting paper concerning the African superplume, and its morphology at depth.

Some of the referencing of the seismological literature is a bit confused and/or out of date, so some of my comments below try to help with that. However, fundamentally, I do not consider this work suitable for publication in Nature Communications. The paper makes overly simplistic assumptions

about the causes of melting (hot plume material into thin spots), and neglects to properly address other issues such as stretching factors and the role of CO₂/metasomes in driving melting. By the end of the paper, I was a bit confused as to precisely what the work genuinely constrains. The discussion is broad ranging, but too speculative in nature. Perhaps most importantly, I don't really see what major step forward is made via this study. The authors need to think hard about what they have genuinely constrained that is new before submitting their work to a longer-format journal.

* Line 36 (and 45) - the references need re-visiting here. Gallacher et al., is a very nice study, but is neither a global or continent-scale model and should be removed from here. Rather it is a study of the uppermost mantle (top 250km only) wavespeed beneath Ethiopia. Similarly, Chang and van der Lee is a very nice study, but not a global or continent-scale model (only Ethiopia and Arabia are imaged). Since the authors are studying the African Superplume, it would seem sensible to focus on seismic studies that image it in its entirety. French and Romanowicz (2015), and Ritsema et al., (2010) seem more relevant here.

* Line 48 - cite French and Romanowicz (2015) here.

* Lines 49-52 - here you introduce the African Superplume as a thermo-chemical anomaly, but without any real discussion of what we know about its composition and thermal structure. It has been argued that the superplume in the lower mantle has a dense chemical layer within a buoyant upward-flowing thermal structure (Ni et al., 2002). Petrological constraints of superplume mantle potential temperature reveal elevated temperatures of +140-170C, consistent with the need for a buoyant component to the superplume.

* Page 3 paragraph 1 - the idea that small upwellings emanate from the African superplume in the lower mantle was first introduced petrologically by Kieffer et al., (2004), then suggested seismically by Bastow et al., (2008). Credit this accordingly, not just to those studies that have championed the idea since.

* Line 60 - I know what you mean when you refer to the magmatic northern half of the EARS, but perhaps tweak this wording somehow? The eastern and western branches of the EAR are often referred to as magmatic and non-magmatic, respectively.

* In lines 63-68 it is stated that:

"Continental lithosphere is inherently heterogeneous, specifically in terms of thickness. The Archean cratonic roots are deep so melts can only be generated beneath pre-existing thinner Karroo and Palaeogene orogenic belts adjacent to cratons⁶. Superplume mantle under the EARS will not generate large volume melting unless the hot plume material flows below areas of thinned lithosphere and P-T is above the solidus. The scattered distribution of rift-related EARS magmatism is explained by a variable lithospheric topography at depths of 100–150 km, which enables channeling of plume material into streams and pools^{2,21}. If this model is correct, then plume mantle of similar composition should underlie most of the EARS."

This requires more detail. Are you saying that all plume material has to do is flow into a region of thinned lithosphere to yield melt? Any old thinned lithosphere, or rapidly thinning lithosphere? Only when stretching factors are taken into account can the effects of decompression melting in response to plate thinning be properly understood. And what of other drivers for melting? CO₂/metasome assisted, whether it be at depth (Rooney et al., 2012), or in the lithosphere (Rooney et al., 2014; Lee et al., 2016)? These issues are critical to discuss before sweeping statements are made concerning plume composition and the drivers of magmatism in East Africa.

* Line 185 - "...and since then, rifting has essentially propagated northwards". Do you mean to say something more like "Oceanic spreading has since developed progressively northwards"?

* Rifting and channeled flow of superplume mantle - when I got to the end of this section, I still couldn't see how the results of the present study really help us understand channel flow at all.

* In the conclusions:

"Thus, the most recent (Mid to Late Miocene) phase of rift-related magmatism along the entire EARS occurs above a low-velocity region containing common superplume mantle, perhaps reflecting changes to the complex plate boundary between the African main continent (Nubia Plate) and Madagascar (Somalia Plate)³⁴, possibly in response to a (global) plate reorganization".

The first part of this is a statement of the obvious. We have known this for an awfully long time. The second part of the text is pure speculation, with links to the current dataset that are not made clear at all.

The final sentence of the conclusions is also acknowledged by the authors to be speculation. I don't think Nature Communications is the place for this kind of thing. The authors need to go back to the drawing board and figure out what they have genuinely contributed to the literature that is new because at present it is not entirely clear.

References cited in this review that don't exist in the MS already

Lee, H., Muirhead, J.D., Fischer, T.P., Ebinger, C.J., Kattenhorn, S.A., Sharp, Z.D. and Kianji, G., 2016. Massive and prolonged deep carbon emissions associated with continental rifting. *Nature Geoscience*, 9(2), p.145.

Ni, S., Tan, E., Gurnis, M. and Helmberger, D., 2002. Sharp sides to the African superplume. *science*, 296(5574), pp.1850-1852.

Ritsema, J., Deuss, A.A., Van Heijst, H.J. and Woodhouse, J.H., 2011. S40RTS: a degree-40 shear-velocity model for the mantle from new Rayleigh wave dispersion, teleseismic traveltime and normal-mode splitting function measurements. *Geophysical Journal International*, 184(3), pp.1223-1236.

Rooney, T.O., Herzberg, C. and Bastow, I.D., 2012. Elevated mantle temperature beneath East Africa. *Geology*, 40(1), pp.27-30.

Rooney, T.O., Nelson, W.R., Dosso, L., Furman, T. and Hanan, B., 2014. The role of continental lithosphere metasomes in the production of HIMU-like magmatism on the northeast African and Arabian plates. *Geology*, 42(5), pp.419-422.

Reviewer #4 (Remarks to the Author):

This is an interesting paper concerning the African superplume, and its morphology at depth.

Some of the referencing of the seismological literature is a bit confused and/or out of date, so some of my comments below try to help with that.

We are gratified that R#4 finds our paper to be interesting and appreciate the help with referencing.

However, fundamentally, I do not consider this work suitable for publication in Nature Communications.

We have improved the clarity and focus our manuscript, deleted unnecessary text and speculative findings and argue why our finding represent a significant advance over Halldórsson et al., (2014) and thus represents a significant advance in our scientific understanding of the East African Rift system.

By the end of the paper, I was a bit confused as to precisely what the work genuinely constrains. The discussion is broad ranging, but too speculative in nature. Perhaps most importantly, I do't really see what major step forward is made via this study. The authors need to think hard about what they have genuinely constrained that is new before submitting their work to a longer-format journal.

As explained above, we have made a considerable number of additions and revisions to clarify our finding and it's significance for our scientific understanding the East African Rift system. We do this by addressing the novelty concerns that our isotopic signature is an incremental advance on the findings of Castillo et al. 2014 and Halldórsson et al 2014. We explain in our revised manuscript why this assumption is incorrect and that our manuscript is a significant advance because we report the first robust, directly measured accurate solid isotopic signature that can be readily detected in EARS lavas. We hope that we have made a strong enough case in our revised manuscript for the reviewer to consider changing her/his mind.

The paper makes overly simplistic assumptions about the causes of melting (hot plume material into thin spots), and neglects to properly address other issues such as stretching factors and the role of CO₂/metasomes (metasomatically enriched domains) in driving melting.

We address these different drivers of melting in our revised manuscript and integrate them into our finding. For conciseness, we address these in more detail as they arise below.

** Line 36 (and 45) - the references need re-visiting here. Gallacher et al., is a very nice study, but is neither a global or continent-scale model and should be removed from here. Rather it is a study of the uppermost mantle (top 250km only) wavespeed beneath Ethiopia. Similarly, Chang an van der Lee is a very nice study, but not a global or continent-scale model (only Ethiopia and Araboia are imaged).*

Since the authors are studying the African Superplume, it would seem sensible to focus on seismic studies that image it in its entirety. French and Romanowicz (2015), and Ritsema et al., (2010) seem more relevant here.

We appreciate the reviewer updating and clarifying the seismological literature. The referencing in our revised paper incorporates the references recommended by all the reviewers. We have not removed Gallacher et al., and Chang and van der Lee because another reviewer argues strongly that our finding is particularly relevant to the shallow part of the low velocity zone under the EARS.

** Line 48 - cite French and Romanowicz (2015) here.*

We now cite this paper again along with French et al., 2013. Both papers were cited in our original submission.

** Lines 49-52 - here you introduce the African Superplume as a thermo-chemical anomaly, but without any real discussion of what we know about its composition and thermal structure. It has been argued that the superplume in the lower mantle has a dense chemical layer within a buoyant upward-flowing thermal structure (Ni et al., 2002). Petrological constraints of superplume mantle potential temperature reveal elevated temperatures of +140-170C, consistent with the need for a buoyant component to the superplume.*

We now discuss that seismic studies have established that the African superplume is a thermos-chemical structure in the lower mantle, raising the possibility that some part of the observed seismic anomaly in the upper mantle is also compositionally.

Our revised text addresses the following points:

- The thermo-chemical nature of the seismic anomaly (superplume) in the lower mantle suggests a dense chemical layer within a buoyant upward-flowing thermal structure (Ni et al., 2002),
- This is most easily explained by recycling of crust into the mantle carrying with it carbonate (CO₂) that can trigger deep melting in both the recycled crust and associated peridotite host (e.g., Dasgupta and Hirschmann, 2006),
- We also discuss that a new generation of petrologic models (Herzberg and Asimow, 2008) show that while the T_p of the East African mantle is characterized by elevated temperatures (+140-170C), consistent with the need for a buoyant component to the superplume. However, while such temperatures are elevated above ambient values, they are nevertheless toward the cooler end of LIPs within the global database,
- Thus, factors other than elevated temperature must be contributing to the exceptionally slow seismic velocity under East Africa,
- We conclude our discussion about the composition and thermal structure of the superplume anomaly by explaining that given the absence of a thermal anomaly of sufficient magnitude, CO₂-assisted melt production in the African superplume likely contributes significantly to the low seismic wave speeds that characterize the East African mantle (Rooney et al., 2012).

** Page 3 paragraph 1 - the idea that small upwellings emanate from the African superplume in the lower mantle was first introduced petrologically by Kieffer et al., (2004), then suggested seismically by Bastow et al., (2008). Credit this accordingly, not just to those studies that have championed the idea since.*

We thank the reviewer for further helping us with the seismological literature. We have included the above references in our revised our manuscript.

** Line 60 - I know what you mean when you refer to the magmatic northern half of the EARs, but perhaps tweak this wording somehow? The eastern and western branches of the EAR are often referred to as magmatic and non-magmatic, respectively.*

We appreciate that this might cause confusion so we now use the term “magma-rich”.

* In lines 63-68 it is stated that:

"Continental lithosphere is inherently heterogeneous, specifically in terms of thickness. The Archean cratonic roots are deep so melts can only be generated beneath pre-existing thinner Karroo and Palaeogene orogenic belts adjacent to cratons⁶. Superplume mantle under the EARS will not generate large volume melting unless the hot plume material flows below areas of thinned lithosphere and P-T is above the solidus. The scattered distribution of rift-related EARS magmatism is explained by a variable lithospheric topography at depths of 100–150 km, which enables channeling of plume material into streams and pools^{2,21}. If this model is correct, then plume mantle of similar composition should underlie most of the EARS."

The above text has been included in response to a comment by another reviewer. The final sentence is no longer included.

This requires more detail. Are you saying that all plume material has to do is flow into a region of thinned lithosphere to yield melt? Any old thinned lithosphere, or rapidly thinning lithosphere? Only when stretching factors are taken into account can the effects of decompression melting in response to plate thinning be properly understood.

We agree that it is essential to provide a more detail and rounded discussion of the drivers of magmatism at magmatic rifted margins other than the widely cited explanation of elevated temperature. We now cite Rooney et al. (2012, 2014) and Lee et al. (2016).

The mechanism we are proposing does not require thinned/rapidly thinning lithosphere. Stretching factors represent a further mechanism for generating large volumes of decompression melt in the asthenosphere as noted Rooney et al., (2012) *"In Ethiopia, plate stretching at different times during rift development has likely produced large volumes of decompression melt in the asthenosphere that markedly lower mantle seismic velocities observed beneath the region (e.g., Bastow et al., 2010; Bastow and Keir, 2011)"*.

Our dataset is consistent with the EARS acting to transfer low-volume melts to the surface via its offshore extensions. The essence of our model is that low degree plume melts bypass contamination and/or dilution by large-volume melting of the subcontinental lithospheric mantle. We do not invoke *large-volume melting* caused by *thinned/rapidly thinning lithosphere*, *'thin spots'*, (significant) *plate stretching* and elevated temperatures.

And what of other drivers for melting? CO₂/metasome assisted, whether it be at depth (Rooney et al., 2012), or in the lithosphere (Rooney et al., 2014; Lee et al., 2016)? The paper makes overly simplistic assumptions about the causes of melting (hot plume material into thin spots), and neglects to properly address other issues such as stretching factors and the role of CO₂/metasomes (metasomatically enriched domains) in driving melting. These issues are critical to discuss before sweeping statements are made concerning plume composition and the drivers of magmatism in East Africa.

Metasome assisted melting

In summary, Rooney et al. 2014 conclude the following:

- *"Melting of metasomatically enriched domains, or metasomes, within the lithospheric mantle provides a viable mechanism for generating the geochemical characteristics of intraplate alkaline basalts."*
- *"The origins and distribution of these metasomes have been attributed to recent enrichment of the lithosphere by a mantle plume or ancient events that occurred during the early evolution of the sub-continental lithosphere mantle. "*
- *"The isotopic characteristics of this component are distinct from the Afar plume mantle (i.e., 'C') source and instead are consistent with the long-term evolution of a lithospheric*

metasome created during a Neoproterozoic subduction event associated with the Pan-African orogeny.’’

The posited metasomatically enriched lithospheric domains, or metasomes, are located within the continental lithosphere and have a distinctly different (HIMU-like) isotopic signature compared to the Afar plume (aka ‘C’). Rooney et al. use this difference to distinguish Neoproterozoic subduction-related ‘metasomes’ from Cenozoic plume (i.e., ‘C’/Afar) lithosphere interaction. This same isotopic distinction rules out contamination of our offshore ‘C’ / Afar OIB samples by, for example, continental metasomes.

Rooney et al conclude that *“The widespread distribution of easily fusible lithospheric metasomes within the continental lithosphere mantle may facilitate magma generation without the need for substantial lithospheric thinning or elevated mantle potential temperatures.*

This mechanism helps to explain the uncontaminated plume signature in offshore OIB lavas without the need for substantial lithospheric thinning, thin spots, (significant) plate stretching or elevated mantle potential temperatures. We thank the reviewer for drawing our attention to this important support for our finding. We also explain that Castillo et al. (2014) invoke a similar mechanism to explain why RVP magmas carry the Pb but not the Sr-Nd aspects of the common C signature.

CO₂ assisted melting

Rooney et al. 2012 suggest the following:

- *“Seismic studies have established that the African superplume is a thermochemical structure in the lower mantle, raising the possibility that some part of the observed seismic anomaly in the upper mantle is also compositionally based.”*
- *“The source of the compositional heterogeneity within the superplume in the lower mantle is likely related to recycled slab materials that have been converted into eclogites and pyroxenites (e.g., Kogiso et al., 2003), consistent with bulk modulus of the African superplume (Tan and Gurnis, 2005; Forte et al., 2010).”*
- *While water is present in the source of ocean island basalts (Dixon et al., 2002), CO₂ is likely the dominant volatile phase during small degrees of partial melting (Dasgupta et al., 2007b).*
- *The recycling of crust into the mantle may carry with it carbonate, and this CO₂ can trigger deep melting in both the recycled crust and associated peridotite host (e.g., Dasgupta and Hirschmann, 2006).*
- *We thus suggest that given the absence of a thermal anomaly of sufficient magnitude, CO₂-assisted melt production in the African superplume likely contributes significantly to the low seismic wave speeds that characterize the East African mantle.*

The notion of volatile-rich (CO₂) carbonatitic plume domains/metasomes involve CO₂ assisted melting and is consistent with the notion that this CO₂ originates from the recycling of crust into the mantle carrying carbonate (CO₂). For example, Rooney et al., 2012 note that *“While water is present in the source of ocean island basalts (Dixon et al., 2002), CO₂ is likely the dominant volatile phase during small degrees of partial melting (Dasgupta et al., 2007b). Carbonate melts are a volumetrically minor component of the mantle, but such melts can cause sufficient dissolution and reprecipitation of olivine to influence the mantle seismic properties (Dasgupta and Hirschmann, 2006).”*

We discuss that the “outgassing” of abundant CO₂ from the carbonatitic plume material at depth can act as the carrier phase for the trace gases, as proposed to explain the decoupling of helium from other geochemical tracers in the Hawaiian plume (e.g., Hofmann et al., 2011).

‘provide strong evidence that significant volumes of CO₂, probably sourced from upper mantle and/or lower crustal magma bodies, are emitted through fault systems positioned away from active volcanic centres in the EAR.’ The authors conclude that *‘Seismicity at depths of 15–30 km implies that extensional faults in this region may penetrate the lower crust. We therefore suggest that CO₂ is transferred from upper-mantle or lower-crustal magma bodies along these deep faults’* Thus, Lee et al. (2016) use CO₂ measurements to infer the transfer of melts along the EARS from upper-mantle or lower-crustal magma bodies via deep faults rather than just volcanic centers such as Nyiragongo. We now cite this as further evidence that low-degree melts are tapped tectonically along the entire the EARS and not just at individual volcanic centres.

Small-scale convection (Rooney et al., 2012)

Small-scale convection is mentioned in passing as another potential driver of melting in Rooney et al. as follows: *‘Therefore, whether the mantle was characterized by elevated potential temperatures (T_P), small-scale convection, or anomalously fertile composition is debated.’*. The other reviewers drew our attention to recent seismological papers showing *‘Small-scale upwellings (about 100 km diameter), with mild excess temperatures, rise from a regional thermal boundary layer at the base of the upper mantle below Afar and west of the Main Ethiopian Rift, and extend throughout the upper mantle (Civiero et al., 2015, 2016)’*. Our revised paper retains the limited discussion about small-scale convection as another driver of melting.

** Line 185 - "...and since then, rifting has essentially propagated northwards". Do you mean to say something more like "Oceanic spreading has since developed progressively northwards"?*

Good point, thanks. We have clarified the text as indicated.

** Rifting and channeled flow of superplume mantle - when I got to the end of this section, I still couldn't see how the results of the present study really help us understand channel flow at all.*

We did not address or invoking channelling of plume material under the EARS. We have removed any mention of ‘channelling’ and emphasis that our meaning is how melts reach the surface.

** In the conclusions:*

"Thus, the most recent (Mid to Late Miocene) phase of rift-related magmatism along the entire EARS occurs above a low-velocity region containing common superplume mantle,.... The first part of this is a statement of the obvious. We have known this for an awfully long time.

We disagree that *‘We have known this for an awfully long time’* that *‘the most recent (Mid to Late Miocene) phase of rift-related magmatism along the entire EARS occurs above a low-velocity region containing common superplume mantle’*,

As we discuss in detail in our revised manuscript there is no robust solid isotopic tracer for the superplume mantle.

perhaps reflecting changes to the complex plate boundary between the African main continent (Nubia Plate) and Madagascar (Somalia Plate)³⁴, possibly in response to a (global) plate reorganization..... The second part of the text is pure speculation, with links to the current dataset that are not made clear at all.....

We have removed this sentence from our revised manuscript. We were considering here the wider implications of our key finding. Rooney et al (2012) address plate stretching factors as follows: ‘*In Ethiopia, plate stretching at different times during rift development has likely produced large volumes of decompression melt in the asthenosphere that markedly lower mantle seismic velocities observed beneath the region (e.g., Bastow et al., 2010; Bastow and Keir, 2011). We consider that it is a reasonable inference that ‘plate stretching at different times during rift development’ is consistent with ‘changes to the complex plate boundary between the African main continent (Nubia Plate) and Madagascar (Somalia Plate)*³⁴?

The final sentence of the conclusions is also acknowledged by the authors to be speculation. I don't think Nature Communications is the place for this kind of thing.

We have removed this sentence.

The authors need to go back to the drawing board and figure out what they have genuinely contributed to the literature that is new because at present it is not entirely clear.

By the end of the paper, I was a bit confused as to precisely what the work genuinely constrains. The discussion is broad ranging, but too speculative in nature. Perhaps most importantly, I do't really see what major step forward is made via this study. The authors need to think hard about what they have genuinely constrained that is new before submitting their work to a longer-format journal.

We have improved the clarity and focus our manuscript by concentrating on the central message about how and why our finding differs from Halldórsson et al., (2014) and so must be considered a significant advance in our scientific understanding of the East African Rift system.

Perhaps the best way to address the above comment/issue is via the re-written first paragraph and summary & conclusions.

First paragraph:

Seismological findings show an increasingly complex scenario of plume upwellings from a deep thermo-chemical anomaly (superplume) beneath the East African Rift System (EARS). It is unclear if these low seismic-velocity anomalies represent a true picture of the posited deep-seated African superplume and its relation with rift-related magmatism along the EARS. Thus, it is essential to find a unique geochemical tracer for establishing where low-velocity upwellings are connected to the deep-seated thermo-chemical anomaly. Here we identify a unique solid superplume isotopic signature (‘C’) in the youngest (after 10 Ma) phase of widespread EARS rift-related magmatism where it extends into the Indian Ocean and the Red Sea. These offshore rock samples provide the first sound evidence that superplume is impacting the magma-poor southern half of the EARS, far from the remarkably low seismic velocities in the magma-rich northern half. Our finding (a) isotopically tracks for the first time superplume mantle the length of Africa from the Indian Ocean to the Red Sea and (b) implies that superplume material is modifying the EARS mantle not only in East African but the length of African from the Red Sea to southern Mozambique. These findings are consistent with seismic images of a single deep-seated superplume. characterized by a more-or-less homogeneous isotopic composition.

Summary & conclusions

- We have sampled basanites-alkaline basalts from one of the small domes on the Mozambique Ridge,

- These OIB domes are located along the offshore extension of the EARS into the Indian Ocean implying that the Mozambique Ridge consists of igneous crust,
- A ~ 7 Ma $^{40}\text{Ar}/^{39}\text{Ar}$ age for these OIB lavas shows that the widespread youngest (after 10 Ma) phase of EARS magmatism extends from the Red Sea as far south as the Mozambique Ridge,
- The OIB lavas express a solid Sr-Nd-Pb isotopic signature ('C'),
- This plume C isotopic signature is sampled also by the offshore extension of EARS into the Red Sea and Somalia Basin,
- No plume C lavas are sampled along the continental EARS due to contamination by large-volume melting of (metasomatised) lithospheric sources,
- We conclude that the solid isotopic superplume signature C is only sampled offshore where the EARS can act to transfer uncontaminated low-volume melts of more fusible (fertile \pm CO_2), relatively cool (TP that ranges from ambient mantle to only 1490 °C), widely distributed plume material to the surface, possibly in association small-scale plume convection,
- These offshore rock samples provide the first sound evidence that superplume material is present in the magma-poor southern half of the EARS, far from the very-low seismic velocities associated with the magma-rich northern half,
- We argue that because the EARS can tap pure plume C lavas wherever the tectonic setting is appropriate then common 'plume C' superplume material must be widely distributed in the upper mantle under the entire EARS from the Mozambique Ridge to the Red Sea-Gulf of Aden,
- This finding implies that superplume material is modifying the EARS mantle not only in East African but the length of African from the Red Sea to southern Mozambique,
- In summary, we have isotopically tracked for the first time superplume mantle the length of Africa from the Indian Ocean to the Red Sea, which is consistent with seismic mantle images of a single deep-seated superplume inference with a more-or-less homogeneous isotopic composition,
- Our finding seems to offers some broader perspectives about the African superplume that raise some interesting questions for future lines of research as follows:
- Evidence that superplume mantle is associated with the magma-poor southern half of the EARS implies that the C mantle is reaching the EARS regardless of whether it is magma-rich or the existence of anomalously low mantle velocities (Fig. 6). This observation is consistent with the suggestion that the markedly low-mantle velocities in the magma-rich northern half of the EARS reflect the thermochemical structure of the superplume due to, for example, recycling of crust into the mantle carrying with it carbonate (CO_2) that can trigger deep melting,
- A further perspective is that it is increasingly more feasible to measure Sr-Nd-Pb signatures in melt inclusions in primary minerals^{Koornneef et al., 2015} that might carry the pure solid superplume C isotopic signature to the surface. These primary melts lock in the C isotopic signature before it can be overwhelmed by contamination from large-volume melting of lithospheric sources. Thus, in due course the superplume 'C' fingerprint could be used to track superplume mantle in contaminate/diluted continental EARS lavas.

References cited in this review that don't exist in the MS already

We thank the reviewer for updating and clarifying the continent-scale and global scale seismological literature encompassing the deep morphology of the African superplume.

Lee, H., Muirhead, J.D., Fischer, T.P., Ebinger, C.J., Kattenhorn, S.A., Sharp, Z.D. and Kianji, G., 2016. Massive and prolonged deep carbon emissions associated with continental

rifting. Nature Geoscience, 9(2), p.145.

Ni, S., Tan, E., Gurnis, M. and Helmberger, D., 2002. Sharp sides to the African superplume. science, 296(5574), pp.1850-1852.

Ritsema, J., Deuss, A.A., Van Heijst, H.J. and Woodhouse, J.H., 2011. S40RTS: a degree-40 shear-velocity model for the mantle from new Rayleigh wave dispersion, teleseismic traveltimes and normal-mode splitting function measurements. Geophysical Journal International, 184(3), pp.1223-1236.

Rooney, T.O., Herzberg, C. and Bastow, I.D., 2012. Elevated mantle temperature beneath East Africa. Geology, 40(1), pp.27-30.

Rooney, T.O., Nelson, W.R., Dosso, L., Furman, T. and Hanan, B., 2014. The role of continental lithosphere metasomes in the production of HIMU-like magmatism on the northeast African and Arabian plates. Geology, 42(5), pp.419-422.

REVIEWERS' COMMENTS:

Reviewer #2 (Remarks to the Author):

After carefully reading the revised manuscript and the answers given by the authors to my questions, I cannot see any reason why the article cannot be published. The authors have done a lot of work improving the manuscript and addressing the concerns of all referees in a convincing fashion. As such, this version is a great improvement over the previous, and I recommend publication.

We are very grateful to R#2 continuing to help us to improve our manuscript. We have improved the clarity and focus our manuscript by concentrating on explaining why our finding differs from Halldórsson et al., (2014) and thus represents a significant advance in our scientific understanding of the East African Rift system. We have now incorporated all of the suggestions and changes mentioned below.

However, here are few points that I strongly advice the author to consider:

The revised title is very similar to one used by Halldorsson et al (GRL 2014). I wonder if the author can be more creative here? Maybe the southernmost extreme of the EARS is relevant to bring up in the title? Or something regarding increased extent of superplume influences under the EARS. Their revised title reads as if this contribution is adding nothing to the work done by Castillo et al and Halldorsson et al, which is not true. Please consider to revise the title. This may potentially improve the future impact of this manuscript.

We thank the reviewer for this an excellent suggestion. We have changed the title to “Superplume mantle tracked isotopically the length of Africa from the Indian Ocean to the Red Sea”. The changed title also addresses the concerns raised by the editor and other reviewers about the novelty of our finding. We also argue much more clearly in the revised manuscript why a solid Sr-Nd-Pb isotopic tracer is a significant advance in our understanding about the EARS and not just an incremental step forward.

The authors state that Young volcanics are now shown in Figure 1 a; I’m sorry, but I do not see this on Figure 1 and there is no Figure 1a? What am I missing here?

We now indicate the locations of the isotopically dated volcanoes and plateaus associated with the most recent phase of rift-related magmatism discussed in the text.

I am sorry, but I still think that the Ar-Ar age spectra should be moved to the supplement? These are analytical details which add little to the main message of this manuscript. In addition, they complicate the presentation of Fig. 2. The focus there should be on the geological characteristics of the region but not the Ar-Ar- age spectra.

We have moved the Ar-Ar age spectra to the Supplementary Information.

•Figure 6b is confusing. On Figure 1, please indicate what area 6b covers.

We now indicate on Figure 1 what area 6b covers.

However, I remain skeptical regarding the importance of Fig. 6 for this manuscript. In my opinion, it does not add much to the manuscript and appears out of place. Why not place it in the supplement? The concept of a seaward extension of the EARS has already been introduced in Fig. 1 and 2. Why repeat this in Fig. 6? This is not clear to me.

We have also moved Figure 6 to the Supplementary Information

•In my opinion, Figure 7 greatly improves the manuscript but Figures 7b and 7c appear to be taken directly from Hansen et al. (EPSL 2012). Not entirely sure, but wonder if Nature Com.

approves this? Why not just refer to the work of Hansen et al. (2012) or introduce a modified version of their figure in Fig. 7c?

We appreciate this comment. But if Nature Com. allows we would prefer to use the figure from Hansen et al. (EPSL 2012). Our thinking is that there is a lot of subtlety (or maybe ambiguity) in the modelled velocities that would be lost otherwise.

REVIEWERS' COMMENTS:

Reviewer #4 (Remarks to the Author):

This is a fine manuscript that should be published. The authors have done a good job of improving this manuscript. Finally, this is my 3rd review of this work and have I nothing to add anymore.